# Test-Time Classifier Adjustment Module for Model-Agnostic Domain Generalization

**Yusuke Iwasawa**
The University of Tokyo
iwasawa@weblab.t.u-tokyo.ac.jp

**Yutaka Matsuo**
The University of Tokyo
matsuo@weblab.t.u-tokyo.ac.jp

## Abstract

This paper presents a new algorithm for domain generalization (DG), *test-time template adjuster (T3A)*, aiming to robustify a model to unknown distribution shift. Unlike existing methods that focus on *training phase*, our method focuses *test phase*, i.e., correcting its prediction by itself during test time. Specifically, T3A adjusts a trained linear classifier (the last layer of deep neural networks) with the following procedure: (1) compute a pseudo-prototype representation for each class using online unlabeled data augmented by the base classifier trained in the source domains, (2) and then classify each sample based on its distance to the pseudo-prototypes. T3A is back-propagation-free and modifies only the linear layer; therefore, the increase in computational cost during inference is negligible and avoids the catastrophic failure might caused by stochastic optimization. Despite its simplicity, T3A can leverage knowledge about the target domain by using off-the-shelf test-time data and improve performance. We tested our method on four domain generalization benchmarks, namely PACS, VLCS, OfficeHome, and TerraIncognita, along with various backbone networks including ResNet18, ResNet50, Big Transfer (BiT), Vision Transformers (ViT), and MLP-Mixer. The results show T3A stably improves performance on unseen domains across choices of backbone networks, and outperforms existing domain generalization methods.

## 1 Introduction

Deep neural networks often fail to generalize to out-of-distribution samples. Accuracy suffers when the model performs under conditions different to those of training, such as variations in light [8], weather [51], object poses [2], textures [16], or object backgrounds [5]. Nevertheless, the model may be deployed to different conditions in practical situations; thus, some countermeasures are needed.

Over the past decade, various studies have focused on training a generalizable model to unseen domains given a dataset consisting of several source domains. This setting is usually denoted as *domain generalization (DG)* [6, 61]. Domain generalization operates under the assumption that one can improve robustness to domain shift by incorporating the structure common to multiple domains. For example, domain-invariant feature learning constrains the representation to be invariant to domain shifts [14, 45, 29]. Other methods use meta-learning [19] to learn how to regularize the model to improve the robustness [28, 4, 31]. However, despite significant work on this front, machine learning systems are still vulnerable to domain shifts even after using the above methods during training. Notably, recent large-scale benchmarks [17] show that many approaches *do not* provide significant improvement compared to simple supervised learning, i.e., empirical risk minimization (ERM), with a proper and practical experimental setup. It suggests that the setup in its current state may be too difficult, and a different approach might be needed.

This paper proposes a method of using additional off-the-shelf data in the DG setup, the unsupervised data available at the *test-time*. Since no data about the target domain is available during *training* in

a DG setup, the existing domain generalization algorithms focus on how to use labeled data from multiple-source domains. However, at test-time the model always has access to test data from the target domain. Although the available data is constrained to be (1) unlabeled and (2) only available online (models can not know all test cases in advance), this data provides clue about the target distribution that is not available during training. It is natural to ask the question: How can we use the off-the-shelf unlabeled data available at *test-time* to increase performance on the target domain?

It is worth emphasizing that our setting is different from the transductive setting [49, 22, 57] where all test cases are known in advance, even though we use test data for adjustment. When testing, the model is usually deployed in some environment, and must work well on various samples that will appear continuously. Similarly, the deployed model usually needs to make correct predictions at that moment; there is no point in going back in time and correcting the predictions. Therefore, it is desirable that adjustment and inference be performed at the same time, not offline after a large amount of data has been accumulated. Looking beyond domain generalization, some recent studies suggest optimizing the model during test time using objective function defined only by unsupervised data (e.g, prediction entropy) [34, 52]. However, updating parameters using stochastic gradient descent (SGD) increases computational costs and harm inference throughput. In addition, data available at test time is limited, and stochastic optimization can lead to catastrophic failure.

To this end, we present *test-time templates adjuster (T3A)*, which adjusts the linear classifier (the last layer of deep neural networks) at test-time. T3A adjusts the weights of the linear classifier as the following optimization-free procedure: (1) create a pseudo-prototype for each class using online unlabeled data and the classifier trained in the source domains, (2) and then classify each sample based on its distance to the pseudo-prototype. This procedure makes the adjusted decision boundary avoid the high-data density region on the *target domain* and reduce the ambiguity (entropy) of predictions, which is known to be connected to classification error [52]. Since T3A does not alter the training phase, it can be used together with existing DG algorithms. Moreover, it can be used together with any classification model since it only adjusts the linear classifier on top of the representations. Some readers may wonder how effective it is to modify only the linear classifier while freezing the representation itself. Later in this paper (Section 3.2), we empirically demonstrate that this modification is indeed beneficial.

We evaluate our method on multiple standard domain generalization benchmarks, namely VLCS [12], PACS [27], OfficeHome [50], and TerraIncognita [5]. We compare our method with (1) various DG algorithms reported in [17] and (2) Tent [52] that minimizes the prediction entropy at test time using SGD. With the standard ResNet50 backbone [18], T3A improves ERM by 1.5 points on average accuracy over four dataset, and outperforms most existing DG algorithms. Furthermore, we evaluated our method with 10 different backbone networks, including residual networks (resnet18 and resnet50), big transfer (BiT-M-R50x3, BiT-M-R101x3, and BiT-M-R152x2 [24]), vision transformers (ViT-B16, ViT-L16, Hybrid ViT [9], DeiT [48]), and MLP-Mixer (Mixer-L16) [47]. The results show that T3A gives a statistically significant performance gain against ERM on all backbone networks.

## 2 Preliminary and Related Work

### 2.1 Domain Generalization

**Problem setup**     Following [6], we assume multiple datasets $D^d = \{(\boldsymbol{x}_i^d, y_i^d)\}_{i=1}^{n_d}$ collected from several different domains $d \in \{1, \cdots, d_{tr}\}$. The dataset $D^d$ from domain $d$ contains identically and independently distributed samples characterized by some probability distribution $P^d(X, Y)$, where $X$ and $Y$ are random variables of input and target, respectively. Then, our goal is to develop a predictor $f(X)$ that performs well on some unseen test domain, which is characterized by a different probability distribution $P(X, Y) \neq P^d(X, Y)$ for all $d \in \{1, \cdots, d_{tr}\}$. Note that one can not assume the target distribution during training, e.g., no data about the target distribution is available at the time. Therefore, the predictor is usually trained on datasets from several source domains. For example, the predictor can be trained by minimizing the empirical risk:

$$\arg\min_{\boldsymbol{\phi}} \frac{1}{d} \sum_{d=1}^{d_{tr}} \frac{1}{n_d} \sum_{i=1}^{n_d} \ell(f(\boldsymbol{x}_i^d), y_i^d), \tag{1}$$

where $\phi$ is the set of the parameters of the function, and $\ell$ is a loss function measuring prediction error. In the rest of this paper, optimizing the predictor with eq. 1 is called ERM. In a real application,

the model will be deployed after training and expected to classify the data in an online manner. For benchmarking the algorithm, given a dataset containing $n_d$ domains, we usually use the leave-one-domain-out procedure, which uses a single domain as a test domain and the others as training domains. The procedure is repeated $n_d$ times, changing the test domain every time.

**Algorithms** A central branch of DG algorithm is domain-invariant feature learning, which explicitly reduces the domain gaps on a space of latent representations. For example, [14] proposed domain-adversarial networks (DANN) which measure the domain gaps via an external domain classifier. CORAL [45] align the second-order statistics of representations among different domains. [29] uses maximum mean discrepancy (MMD) to measure the domain gap. Many extension have been proposed [33, 1, 21], but they are all the same in that they enhance domain invariance. Another branch are meta-learning-based methods [28, 4, 31], which divide the available domains into meta-train-domains and meta-test-domain and regulate the model trained in meta-train-domains to be useful for the meta-test-domain. Invariant risks minimization [3] regularizes ERM with a gradient normalization penalty over a dummy classifier. Several studies propose to augment the data using mixup [59] between two source domains, which implicitly enhances invariance to domain shifts [56, 58, 53].

**Key differences** As briefly mentioned above, existing domain generalization algorithms focus on the *training phase*; how to regularize the predictor using the knowledge from multiple source domains. Our work focuses on the *test phase*; how to adjust the model using online and unlabeled data, which can characterize the target distribution. Note that proposed method works fully online; It does not require access to offline unlabeled data, and therefore can be compared fairly with existing DG methods

## 2.2 Other Related Work

**Unsupervised domain adaptation.** Our work is related to unsupervised domain adaptation (UDA) [37, 38, 55] as both methods aim to adapt a model given unsupervised data. However, our work primarily differs from UDA in that UDA focuses on adapting during training, while we focus on adapting during testing. In other words, UDA assumes we can access labeled data from the source domain and (unlabeled) data from the target domain at the same time, which is not always possible.

**Source-free domain adaptation.** Among them, recent "source-free" setups are particularly similar to our setting [30, 25, 34]. In these setups, source data is not needed during the adaptation phase, and the model is adapted using the unlabeled data solely from the target domain. However, this adaptation is usually made in an offline manner, i.e., these source-free methods optimize offline with multiple losses for multiple epochs. Our method adjusts the classifier in an online-manner, and therefore is suitable for a domain generalization setup where the trained model is assumed to be deployed.

**Test-time adaptation.** Regarding the problem setup, our work is most closely related to test-time adaptation [52] or test-time training [46]. Notably, [52] proposes fully test-time adaptation, which modulates the BN parameter by minimizing the prediction entropy using stochastic gradient descent. The concept of test-time adaptation is very similar to our work and can be used in domain generalization, however, it has not been fully investigated under the domain generalization setup. Moreover, recent architectures do not employ batch normalization either on pre-training (mainly to avoid the large memory usage required by BN) or fine-tuning phase (for improving performance). Besides, minimizing the prediction entropy using SGD could lead to trivial solutions, such as being biased to predict only a particular class. In comparison, (1) our method can be used together with any classification models since it only adjusts the linear classifier on the top of the representations, (2) our method alleviates catastrophic forgetting due to not using SGD during test-time.

**Prototypical networks.** Before deep learning become popular, the prototype-based classifier is well-investigated in the context of semi-supervised learning [10], continual learning [35, 39], and few-shot learning [44]. Our work is most similar to prototypical networks [44], which combine prototypical classifier and deep neural networks as with our method. However, the use-cases mentioned above of prototypical networks assume access to a few labeled data from the same domain, which differs in our case. To handle the difference, we combine prototypical networks with pseudo-labeling techniques, which are often used in domain transfer literature [42]. Besides, connection to entropy minimization is a new perspective introduced in this paper.

**Algorithm 1** Algorithm of T3A for prediction.

---

**Input:** Feature extractor $f_{\boldsymbol{\theta}}$, the batch of input $\mathbb{B}$, and support sets $\mathbb{S}^k$ available at this point.
**Output:** Prediction for all $\boldsymbol{x} \in \mathbb{B}$, where $\boldsymbol{x} \sim P(\boldsymbol{X})$.
   # Step1. Adjust the template for each class using the $\mathbb{B}$.
   **for** $\boldsymbol{x} \in \mathbb{B}$ **do**
      $\hat{y} = \arg\max q_{\boldsymbol{\omega}}(Y = y_k | f_{\boldsymbol{\theta}}(\boldsymbol{x}))$ (eq. 2)
      $\mathbb{S}^k = \mathbb{S}^k \cup \{\frac{f_{\boldsymbol{\theta}}(\boldsymbol{x})}{\|f_{\boldsymbol{\theta}}(\boldsymbol{x})\|}\}$ for $y^k = \hat{y}$ (eq. 3)
   **end for**
   Filter support sets with eq. 6
   # Step2. Predict based on the distance between the adjusted template.
   **return** $\arg\max \gamma(Y = y_k | f_{\boldsymbol{\theta}}(\boldsymbol{x}))$ for all $\boldsymbol{x} \in \mathbb{B}$ (eq. 4)

---

# 3 Proposal: Optimization-Free Test-Time Classifier Adjustment Module

We propose to replace the output layer of the predictor trained on source-domains (i.e., linear classifier) to a *pseudo-prototypical classifier*, whose prototype features are adjusted during test time while fixing the features already trained on the source domains. We call our method *T3A*, for Test-Time Templates Adjuster. We first explain the detailed algorithm (Section 3.1), and explain how and why it works (Section 3.2). Algorithm 1 outlines the procedure.

## 3.1 Algorithm

We assume the predictor is deep neural networks (DNN) obtained by some learning algorithm (e.g., ERM, DANN, CORAL, etc.) using data from source domains. For convenience, the entire DNN is divided into a linear classifier $q_{\boldsymbol{\omega}}$ for the last layer and a feature extractor $f_{\boldsymbol{\theta}}$ for the rest, where $\boldsymbol{\omega}$ and $\boldsymbol{\theta}$ are the parameters of neural networks. In usual domain generalization setup, $f_{\boldsymbol{\theta}}$ and $q_{\boldsymbol{\omega}}$ are used to predict data from the test domain. For new data $\boldsymbol{x}$, the prediction is given by taking the argmax over the following approximated probability distribution:

$$\arg\max_{y_k} q_{\boldsymbol{\omega}}(Y = y_k | f_{\boldsymbol{\theta}}(\boldsymbol{x})) = \frac{\exp(\boldsymbol{z} \cdot \boldsymbol{\omega}^k)}{\sum_j \exp(\boldsymbol{z} \cdot \boldsymbol{\omega}^j)}, \tag{2}$$

where $\boldsymbol{z} = f_{\boldsymbol{\theta}}(\boldsymbol{x})$, $\boldsymbol{\omega}^k \in \mathbb{R}^{z_{dim}}$ is the $k$-th element of the weights matrix in $\boldsymbol{\omega}$, and $z_{dim}$ is the dimension of $\boldsymbol{z}$, which depends on the feature extractor $f_{\boldsymbol{\theta}}$. In this prediction, $\boldsymbol{\omega}^k$ works as the template of representation for the class $k$, and prediction is done by measuring the distance (dot product) between the template and the representation of the input data. Since this template was trained in the source domain, there is no guarantee that it will be a good template in the target domain.

T3A adjusts the templates during test-time. Assume we have (batch of) test-data $\boldsymbol{x}$ at time $t$ drawn from target distribution $P(\boldsymbol{X}) \neq P^d(\boldsymbol{X})$ for all $d \in \{1, \cdots, d_{tr}\}$. As each prototype should be related to some class, we first augment the input data $\boldsymbol{x}$ via pseudo label $\hat{y}$, which is obtained via eq. 2. Then, we update a support set $\mathbb{S}_t^k$ as follows:

$$\mathbb{S}_t^k = \begin{cases} \mathbb{S}_{t-1}^k \cup \{\frac{f_{\boldsymbol{\theta}}(\boldsymbol{x})}{\|f_{\boldsymbol{\theta}}(\boldsymbol{x})\|}\} & \text{if } \hat{y} = y^k \\ \mathbb{S}_{t-1}^k & \text{else,} \end{cases} \tag{3}$$

where $\|\boldsymbol{a}\|$ represents the L2 norm of the vector $\boldsymbol{a}$ and $\mathbb{S}_0^k = \{\frac{\boldsymbol{\omega}^k}{\|\boldsymbol{\omega}^k\|}\}$. If the input data contains multiple samples at the same time (e.g., a batch of data), the above procedure is repeated for each sample in the batch. Then, prediction is done by taking the argmax over the following adjusted probability distribution:

$$\arg\max_{y_k} \gamma_{\boldsymbol{c}}(Y = y_k | f_{\boldsymbol{\theta}}(\boldsymbol{x})) = \frac{\exp(\boldsymbol{z} \cdot \boldsymbol{c}^k)}{\sum_j \exp(\boldsymbol{z} \cdot \boldsymbol{c}^j)}, \tag{4}$$

where $\boldsymbol{c}^k$ are the centroids of $\mathbb{S}^k$:

$$\boldsymbol{c}^k = \frac{1}{|\mathbb{S}^k|} \sum_{\boldsymbol{z} \in \mathbb{S}_t^k} \boldsymbol{z}. \tag{5}$$

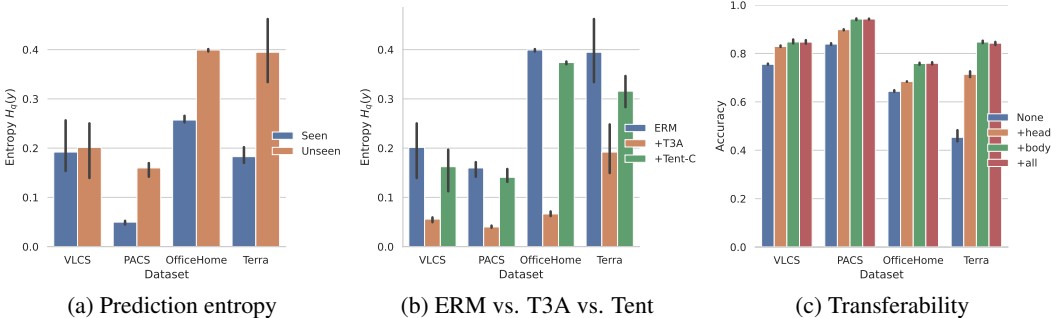

| (a) Prediction entropy | (b) ERM vs. T3A vs. Tent | (c) Transferability |

Figure 1: Pre-experimental results. (a) Comparing the entropy of predictions (with ResNet50) on the source domains and the target domain. The domain shift increases entropy. (b) T3A effectively reduces entropy. (c) Transferability of each components of the model trained by ERM in each dataset.

Note that, this procedure will be repeated each time new data arrives, without discarding the support sets, which may make infeasible to retain all past data as in eq. 3. Also, some pseudo-labels are assigned to the wrong class, and using this data is not desirable as it adds noise to the templates and may deteriorate performance. To avoid this issue, we use the prediction entropy $H_{\boldsymbol{\omega}}(\hat{Y}|\boldsymbol{z}) = -\sum_k q_{\boldsymbol{\omega}}(\hat{Y} = y^k|\boldsymbol{z}) \log q_{\boldsymbol{\omega}}(\hat{Y} = y^k|\boldsymbol{z})$ to filter unreliable pseudo-labeled data. Specifically, before making a prediction using eq. 4, only a part of the support set is restored as follows:

$$\mathbb{S}_t^k = \{\boldsymbol{z} \mid \boldsymbol{z} \in \mathbb{S}_t^k, H_{\omega}(\hat{Y}|\boldsymbol{z}) \leq \alpha^k\}, \tag{6}$$

where $\alpha^k$ is the $M$-th largest entropy of the support set $\mathbb{S}_t^k$ ($M$ is a hyperparameter).

### 3.2 Remarks

**Remark 1: T3A implicitly reduces prediction entropy.** As prior works suggest [52], the prediction entropy is often related to an error, as more confident predictions tend to be more correct. Figure 1-a shows that the prediction entropy also characterizes the difficulty in DG setup; entropy in the unseen domain tends to be greater than entropy in the seen domains. To be more specific, we first trained ResNet50 on the source domain by ERM. The training was done in leave-one-domain-out manner, and we conducted three experiments with a different seed each time. We used four standard datasets in domain generalization (VLCS [12], PACS [27], OfficeHome [50], and TerraIncognita [5]). We used the implementation of `DomainBed` [17], and used the default hyper-parameters for pre-training on source-domains and fine-tuning on a target domain: namely, we use Adam [23] with a learning rate of 5e-5 for optimization and use a batch size of 32 with no dropout or weight decay.

With this in mind, existing studies have modified the model parameters to explicitly reduce entropy. Although the proposed method does not explicitly reduce entropy, it has the effect of *implicitly reducing it*. This is because the proposed method uses a template updated with samples from the target distribution, which provides a decision boundary that avoids the dense parts of the target distribution. Figure 1-b compares the prediction entropy on the target domain among (a) ERM without test-time modulation, (b) T3A, (c) Tent-C, which updates the classifier to minimize entropy. The results show that T3A can effectively reduce entropy without using online optimization. Note that changing a hyper-parameter (such as learning rate) might change the results for Tent-C, but it may corrupt the entire classifier. See Section 4 for the hyperparameter selection.

**Remark 2: T3A is computationally light.** Unlike Tent, our method does not use SGD. Besides, the representations are fixed, and it is not necessary to repeat the forward propagation of the feature extractor. The only computational overhead is the cost of one forward propagation of the last linear layer, which is usually negligible compared to the forward and back propagation of feature extractors. Since it is not desirable to reduce throughput when considering online prediction, the proposed method is suitable in this respect as well.

**Remark 3: Adjusting the linear classifier can significantly improve performance.** Some readers may wonder how effective it is to modify only the linear classifier while freezing the representation. To answer this question, we compare the DG performance (None) before fine-tuning,

(head) after fine-tuning only $q_{\omega}$, (body) after fine-tuning only $f_{\theta}$, and (all) after fine-tuning the entire network (Figure 1-c). Each block corresponds to a different dataset, and each color represents a different fine-tuning strategy. The results show that fine-tuning only the classifier often significantly improves performance. For example, in VLCS, the average performance score jumps from 72.5 to 82.9, which is close to the 84.7 obtained when the entire network is fine-tuned. The performance gain differs for each dataset, but the tendency is generally the same. In addition, this tendency was the same for other backbone networks including BiT, ViT, and Mixer (see Appendix B). These results indicate that adjusting only the linear classifier can significantly improve performance in various configurations. Note that the number of parameters of the linear layer is much smaller than those of the feature extractor in standard network architectures.

## 4 Experiment

We evaluate T3A on four standard domain generalization benchmarks, namely VLCS [12], PACS [27], OfficeHome [50], and TerraIncognita [5]. Our implementation uses the `DomainBed` library [17][1]. We modify `DomainBed` (1) to use various backbone networks using the `timm` library [54][2], and (2) to implement test-time adaptation algorithms (ours and Tent). For Tent, we used the original implementation[3]. We run our experiments mainly on cloud V100x4 or A100x8 instances, depending on the memory usage of the backbone networks. See Appendix A for more information, including licensing information and total amount of compute.

**Datasets.** VLCS [12] comprises four photographic datasets $d \in \{$Caltech101[13], LabelMe[40], SUN09[7], VOC2007[11]$\}$, containing $10,729$ examples of 5 classes. PCAS [27] comprises four domains $d \in \{$art, cartoons, photos, sketches$\}$, containing $9,991$ examples of 7 classes. OfficeHome [50] includes domains $d \in \{$art, clipart, product, real$\}$, containing $15,588$ examples and 65 classes. TerraIncognita [5] includes photo of wild animals taken by camera at different locations. Following [17], we used datasets of $d \in \{$L100, L38, L43, L46$\}$, containing $24,788$ examples and 10 classes.

**Backbone networks.** For the main experiments, we use residual networks with 50 layers (ResNet50), which was the default setting of the prior studies. In addition, we tested our algorithms on 10 different pre-trained models: residual networks with different layers (ResNet18 and ResNet50), Big Transfer [24] with different layers (BiT-M-R50x3, BiT-M-R101x3, and BiT-M-R152x2 ), Vision Transformers [9] with variations (ViT-B16, ViT-L16, HViT, which uses ResNet50 as patch embedding of ViT, DeiT [48]), and MLP-Mixer (Mixer-L16) [47].

**Baselines.** We compare our method to domain generalization algorithms and test-time adaptation algorithms. For domain generalization algorithms, we mainly compared with the results reported in [17]. These results include the following algorithms: Empirical Risk Minimization (ERM), Group Distributionally Robust Optimization (GroupDRO) [41], Inter-domain Mixup (Mixup) [56, 58, 53], Meta-Learning for Domain Generalization (MLDG) [28], DomainAdversarial Neural Networks (DANN) [15], Class-conditional DANN (C-DANN) [32], Deep CORrelation ALignment (CORAL) [45], Maximum Mean Discrepancy (MMD) [29], Invariant Risk Minimization (IRM) [3], Adaptive Risk Minimization (ARM) [60], Marginal Transfer Learning (MTL) [6] Style-Agnostic Networks (SagNet) [36], and Representation Self Challenging (RSC) [20].

We also compared our method with existing test-time adaptation methods. Note that we can not simply use BN-based methods (including Tent [52], which is the most up-to-date method) on the DG setup because [17] omit the BN layer from pre-trained ResNet when fine-tuning on source domains. Besides, several backbone networks evaluated in our paper do not contain BN from the beginning. Therefore, we first tested two slightly modified versions of Tent on standard DG setup (Table 1, Table 2, and Figure 2). Specifically, Tent-C modulates the entire classifier to reduce prediction entropy. Tent-BN adds one BN layer just before the linear classifier and then modulates BN's normalization and transformation parameters. We then compare T3A with other test-time adaptation methods (including SHOT [34], pseudo labeling (PL) [26], Tent-Full [52], BN-Norm [43]) using ResNet18 and ResNet50 without removing batch normalization layer (Table 3).

---

[1]https://github.com/facebookresearch/DomainBed
[2]https://github.com/rwightman/pytorch-image-models
[3]https://github.com/DequanWang/tent

Table 1: Domain generalization accuracy for all datasets and algorithms. Bold type indicates performance improvement from the base model, and * indicates statistical significance in one-sided paired t-test (** indicates $p \leq 0.01$, * indicates $p \leq 0.05$).

| Algorithm | VLCS | PACS | OfficeHome | Terra | Avg |
|---|---|---|---|---|---|
| ERM | $77.5 \pm 0.4$ | $85.5 \pm 0.2$ | $66.5 \pm 0.3$ | $46.1 \pm 1.8$ | 69.0 |
| IRM | $78.5 \pm 0.5$ | $83.5 \pm 0.8$ | $64.3 \pm 2.2$ | $47.6 \pm 0.8$ | 68.5 |
| GroupDRO | $76.7 \pm 0.6$ | $84.4 \pm 0.8$ | $66.0 \pm 0.7$ | $43.2 \pm 1.1$ | 67.6 |
| Mixup | $77.4 \pm 0.6$ | $84.6 \pm 0.6$ | $68.1 \pm 0.3$ | $47.9 \pm 0.8$ | 69.5 |
| MLDG | $77.2 \pm 0.4$ | $84.9 \pm 1.0$ | $66.8 \pm 0.6$ | $47.7 \pm 0.9$ | 69.2 |
| CORAL | $78.8 \pm 0.6$ | $86.2 \pm 0.3$ | $68.7 \pm 0.3$ | $47.6 \pm 1.0$ | 70.3 |
| MMD | $77.5 \pm 0.9$ | $84.6 \pm 0.5$ | $66.3 \pm 0.1$ | $42.2 \pm 1.6$ | 67.7 |
| DANN | $78.6 \pm 0.4$ | $83.6 \pm 0.4$ | $65.9 \pm 0.6$ | $46.7 \pm 0.5$ | 68.7 |
| CDANN | $77.5 \pm 0.1$ | $82.6 \pm 0.9$ | $65.8 \pm 1.3$ | $45.8 \pm 1.6$ | 67.9 |
| MTL | $77.2 \pm 0.4$ | $84.6 \pm 0.5$ | $66.4 \pm 0.5$ | $45.6 \pm 1.2$ | 68.5 |
| SagNet | $77.8 \pm 0.5$ | $86.3 \pm 0.2$ | $68.1 \pm 0.1$ | $48.6 \pm 1.0$ | 70.2 |
| ARM | $77.6 \pm 0.3$ | $85.1 \pm 0.4$ | $64.8 \pm 0.3$ | $45.5 \pm 0.3$ | 68.3 |
| VREx | $78.3 \pm 0.2$ | $84.9 \pm 0.6$ | $66.4 \pm 0.6$ | $46.4 \pm 0.6$ | 69.0 |
| RSC | $77.1 \pm 0.5$ | $85.2 \pm 0.9$ | $65.5 \pm 0.9$ | $46.6 \pm 1.0$ | 68.6 |
| ERM$^\dagger$ | $77.7 \pm 0.1$ | $83.6 \pm 0.9$ | $66.4 \pm 0.3$ | $46.5 \pm 0.3$ | 68.6 |
| +T3A (Ours) | $\mathbf{80.0} \pm 0.2$ | $\mathbf{85.3} \pm 0.6$ | $\mathbf{68.3} \pm 0.1$ | $\mathbf{47.0} \pm 0.6$ | $\mathbf{70.1}^{**}$ |
| +Tent-BN | $68.2 \pm 0.2$ | $\mathbf{84.8} \pm 0.5$ | $\mathbf{67.0} \pm 0.4$ | $44.7 \pm 0.3$ | 66.2 |
| +Tent-C | $77.0 \pm 0.4$ | $82.3 \pm 1.2$ | $65.7 \pm 0.2$ | $45.5 \pm 0.4$ | 67.6 |
| CORAL$^\dagger$ | $78.6 \pm 0.5$ | $84.2 \pm 0.3$ | $68.3 \pm 0.1$ | $48.1 \pm 1.3$ | 69.8 |
| +T3A (Ours) | $\mathbf{79.5} \pm 0.5$ | $\mathbf{85.6} \pm 0.2$ | $\mathbf{69.2} \pm 0.2$ | $47.3 \pm 0.7$ | $\mathbf{70.4}^{*}$ |
| +Tent-BN | $71.4 \pm 0.7$ | $\mathbf{85.6} \pm 0.2$ | $\mathbf{69.2} \pm 0.2$ | $46.5 \pm 0.5$ | 68.2 |
| +Tent-C | $78.1 \pm 0.5$ | $83.7 \pm 0.4$ | $68.2 \pm 0.1$ | $47.8 \pm 1.1$ | 69.5 |

**Hyperparameters and model selection.** As [17] claimed, model selection is not trivial in DG and significantly affects performance. We used standard training-domain validation for selecting hyperparameters, which uses the subset of each training domain to choose a model. As reported in [17], we split the data from each domain into $80\%$ and $20\%$ splits and use larger splits for training and smaller splits to select hyperparameters. Following [17], we conduct a random search of 20 trials over a joint distribution of all hyperparameters to train the base model (see Appendix A.4).

In addition, T3A has one hyperparameter $M$ for deciding the number of supports to restore, and Tent has two primary hyperparameters: $\beta$ for multiplying the base learning rate (used for the base model) and $\gamma$ for the number of iterations per adaptation. It is worth emphasizing that *these parameters should be selected before the deployment, i.e., before accessing the test data*. We simply selected these hyperparameters by the average accuracy in the training-domain validation data when using these adjustment modules. Specifically, we tested $M \in \{1, 5, 20, 50, 100, \text{N/A}\}$ for T3A, where N/A means restoring all samples, and combination of $\beta \in \{0.1, 1.0, 10.0\}$ and $\gamma \in \{1, 3\}$ for Tent.

### 4.1 Results

Table 1 summarizes the results when ResNet50 is used as the backbone network. The first block (from ERM to RSC) is the value taken from [17]. The lines labeled ERM$^\dagger$ and CORAL$^\dagger$ are the scores reproduced in our environments. The proposed method and Tent are based on this reproduced model. Figure 2 shows the distribution of performance improvement by the proposed method for models trained with different hyperparameters ($20 \times 3$ for each test environment). In addition, Table 2 show the DG accuracy with 10 different backbones. Note that this experiment is conducted only on the default hyperparameter of ERM. Every number we report is a mean and standard error over three repetitions with different weight initialization, and dataset splits.

**T3A stably improves the performance of the base model.** The second block of Table 1 shows that T3A stably improves the performance of the ERM model. Specifically, the proposed method improves 2.3 points, 2.0 points, 1.9 points, and 0.5 points for each dataset respectively. The average

Table 2: Domain generalization accuracy with different backbone networks. T3A increases the performance agnostic to backbone networks. Note that, this experiments is conducted only on the default hyperparameters of ERM. Bold type indicates performance improvement, and * indicates statistical significance in paired t-test (** indicates $p \le 0.01$, * indicates $p \le 0.05$).

| Models | VLCS | PACS | OfficeHome | Terra | Avg |
|---|---|---|---|---|---|
| resnet18 | $73.2 \pm 0.9$ | $80.3 \pm 0.4$ | $55.7 \pm 0.2$ | $40.7 \pm 0.3$ | 62.5 |
| +T3A | $\mathbf{76.5 \pm 0.9}$ | $\mathbf{81.7 \pm 0.1}$ | $\mathbf{57.0 \pm 0.4}$ | $\mathbf{41.6 \pm 0.5}$ | 64.2* |
| resnet50 | $75.5 \pm 0.1$ | $83.9 \pm 0.2$ | $64.4 \pm 0.2$ | $45.4 \pm 1.2$ | 67.3 |
| +T3A | $\mathbf{78.3 \pm 0.7}$ | $\mathbf{84.5 \pm 0.3}$ | $\mathbf{66.5 \pm 0.2}$ | $\mathbf{45.9 \pm 0.5}$ | 68.8* |
| BiT-M-R50x3 | $76.7 \pm 0.1$ | $84.4 \pm 1.2$ | $69.2 \pm 0.6$ | $52.5 \pm 0.3$ | 70.7 |
| +T3A | $\mathbf{79.7 \pm 0.3}$ | $\mathbf{85.4 \pm 0.9}$ | $\mathbf{71.7 \pm 0.6}$ | $52.2 \pm 0.6$ | 72.3* |
| BiT-M-R101x3 | $75.0 \pm 0.6$ | $84.0 \pm 0.7$ | $67.7 \pm 0.5$ | $47.8 \pm 0.8$ | 68.6 |
| +T3A | $\mathbf{78.6 \pm 0.4}$ | $\mathbf{85.4 \pm 0.5}$ | $\mathbf{69.9 \pm 0.4}$ | $\mathbf{48.1 \pm 0.8}$ | 70.5* |
| BiT-M-R152x2 | $76.7 \pm 0.3$ | $85.2 \pm 0.1$ | $71.3 \pm 0.6$ | $51.4 \pm 0.6$ | 71.1 |
| +T3A | $\mathbf{79.1 \pm 0.4}$ | $\mathbf{86.4 \pm 0.1}$ | $\mathbf{73.2 \pm 0.5}$ | $50.9 \pm 0.7$ | 72.4* |
| ViT-B16 | $79.2 \pm 0.3$ | $85.7 \pm 0.1$ | $78.4 \pm 0.3$ | $41.8 \pm 0.6$ | 71.3 |
| +T3A | $\mathbf{80.2 \pm 0.4}$ | $\mathbf{86.0 \pm 0.1}$ | $\mathbf{78.9 \pm 0.3}$ | $\mathbf{42.5 \pm 0.7}$ | 71.9* |
| ViT-L16 | $78.2 \pm 0.5$ | $84.6 \pm 0.5$ | $78.0 \pm 0.1$ | $42.7 \pm 1.9$ | 70.9 |
| +T3A | $\mathbf{79.0 \pm 0.6}$ | $\mathbf{85.5 \pm 0.6}$ | $\mathbf{78.7 \pm 0.2}$ | $\mathbf{45.3 \pm 0.4}$ | 72.1** |
| DeiT | $79.3 \pm 0.4$ | $87.8 \pm 0.5$ | $76.6 \pm 0.3$ | $50.0 \pm 0.2$ | 73.4 |
| +T3A | $\mathbf{81.3 \pm 0.4}$ | $\mathbf{89.5 \pm 0.4}$ | $\mathbf{78.3 \pm 0.2}$ | $\mathbf{50.1 \pm 0.2}$ | 74.8* |
| HViT | $79.2 \pm 0.5$ | $89.7 \pm 0.4$ | $80.0 \pm 0.2$ | $51.4 \pm 0.9$ | 75.1 |
| +T3A | $\mathbf{81.0 \pm 0.1}$ | $\mathbf{90.4 \pm 0.5}$ | $\mathbf{80.5 \pm 0.2}$ | $\mathbf{52.3 \pm 1.0}$ | 76.1* |
| Mixer-L16 | $76.4 \pm 0.2$ | $81.3 \pm 1.0$ | $69.4 \pm 1.6$ | $37.1 \pm 0.4$ | 66.1 |
| +T3A | $\mathbf{80.3 \pm 0.3}$ | $\mathbf{83.0 \pm 0.8}$ | $\mathbf{72.3 \pm 1.8}$ | $\mathbf{37.5 \pm 0.8}$ | 68.3* |

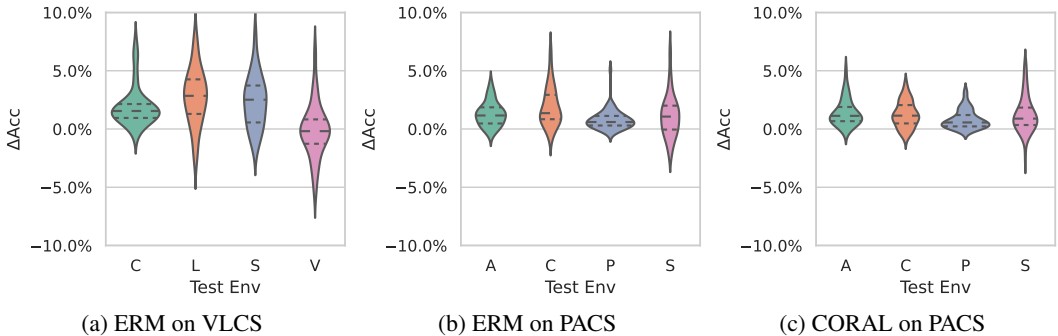

(a) ERM on VLCS     (b) ERM on PACS     (c) CORAL on PACS

Figure 2: Distribution of performance improvements by the proposed method for models trained with different hyperparameters ($20 \times 3$ for each test environment). Dashed line in each violin plot represents the quartile of the distribution.

improvement from ERM is 1.5 points. For clarification, paired t-test was performed using 48 paired data (4 datasets, 4 test domains, 3 different seeds). As a results, the difference is statistically significant ($p \le 0.01$). Note that, Tent-BN improves the performance in PACS and OfficeHome, but it is not stable may be due to the failure of the optimization. Tent-C never improve the performance.

In addition, Fig. 2 show that the third-quarter quantile of the improvement range is often more than 0, which means the proposed method is also robust to the hyperparameters of the base model. Furthermore, Table 2 shows that the proposed method can also improve the performance of more sophisticated backbone networks. For example, the proposed method improves performance by an average of 1.0 points for HViT, which achieved the best performance of all backbones. The

improvement by the proposed method is statistically significant ($p \leq 0.05$ for ViT-L16, $p \leq 0.01$ for other backbones) with the one-side paired t-test.

**T3A outperforms most existing DG algorithms.** For example, in VLCS, the proposed method achieves $80.0\%$, which is significantly better than the prior best-reported score, $78.8\%$. Similarly, it is $85.3\%$ for PACS, $68.3\%$ (best) for OfficeHome and $47.0\%$ for TerraIncognita, and $70.1\%$ in average (third best). Note that the scores we reproduce are much worse than the reported scores in PACS, which reduces the average performance. In PACS, the improvement from ERM by the proposed method is $1.7$ points, which is the most significant improvement from any reported value.

For a more accurate comparison, we also examined whether the proposed method would improve the performance of CORAL, which had achieved the best performance in existing reports. As shown in the third block of Table 1, the proposed method can also improve the performance of CORAL. The average improvement is $0.6$ points. Average performance is $70.4\%$, which is the best among all algorithms. Note that, similar to ERM, the reproduced score and the reported score are slightly different (and the score in PACS is particularly low).

**T3A outperforms existing test-time adaptation methods.** We further compare T3A with other test-time adaptation methods when backbone networks employ the BN layer so that we can make a fair comparison with the prior BN-based methods [43, 52]. We used ResNet18 w/BN and ResNet50 w/BN as backbone networks. Note that the results of ResNet in Table 1 and Table 2 do not use the BN layer since it is the default option in [17]. Therefore, the results below are not directly comparable to the DG method shown in Table 1 and Table 2 of the current manuscript.

We tested the following six baselines in addition to T3A, Tent-BN, and Tent-C. (1) Tent-Full updates BN statistics and transformations, which is the same as the original proposal [52]. (2) BN-Norm update BN statistics but fixes transformations parameters [43]. (3) PL (Pseudo Label) [26] updates entire networks by minimizing the cross-entropy between prediction and the pseudo label. Following [52], we assign the pseudo label if the predictions are over a threshold (0.9 in our experiment). (4) PL-C updates the linear classifier by minimizing the above-mentioned pseudo-label loss. (5) SHOT [34] updates feature extractor to minimize entropy, diversity regularizer, and pseudo-label loss. While [34] originally proposed SHOT in the context of source-free domain adaptation (offline adaptation setup), the method itself can be transferred to our setup. Note that the original SHOT uses the label-smoothing when training on the source domain. However, we focus on the adaptation method, and therefore the source model is the same as the other baselines for the fair comparison. (6) SHOT-IM [34] updates the feature extractor to minimize entropy and the diversity regularizer.

In summary, (1) all baseline except BN-Norm and T3A use stochastic optimization during test-time, which is not desirable since it may cause catastrophic failure and must increase computational costs. (2) Tent-Full and BN-Norm are powerful yet constrained to be applicable only if the architecture uses BN.

Table 3 compares results under the training-domain model selection as with Table 1 and Table 2. For clarification, we also compare performance under the oracle model selection (28 in Appendix C ), where one can use the validation set on the target domain (20% of all data as described above). We can make the following observations. (1) The proposed method still outperformed all baselines in both backbone networks. Among baselines, only Tent-Full and PL-C perform better than None (w/o adaptation) on average. (2) When we select the hyper-parameters with a test-domain validation set, Tent-Full gives comparable performances with the proposed method. In addition, compared to T3A and PL-C, the T3A performs better under both model selection strategies. These results clarify the difficulty of model selection in optimization-based methods and the merit of the proposed optimization-free approach in this setup. (3) Updating feature extractor (or the large portion of the parameters) does not work well in general, while it is common in SFDA (offline) setup. The results suggest that we need different treatments on online and offline setup.

## 5  Discussion and Conclusion

This paper presents a new domain generalization algorithm, T3A, which adjusts its predictions during test-time by itself. We show that T3A reduces prediction entropy (Fig. 1-c), and more importantly, generalization error on unseen domain (Table 1). T3A can adapt different domain generalization

Table 3: Comparison of our method and existing test-time adaptation methods. Unlike Table 1 and Table 2, we used ResNet18 and ResNet50 *without removing batch normalization layer* as backbone networks. As with Table 2, this experiments is conducted only on the default hyperparameters of ERM. Bold type indicates performance improvement, and * indicates statistical significance in paired t-test (* indicates $p \leq 0.05$).

| Models | VLCS | PACS | OfficeHome | Terra | Avg |
|---|---|---|---|---|---|
| resnet18 w/ BN | $73.0 \pm 0.6$ | $79.5 \pm 0.4$ | $61.8 \pm 0.3$ | $41.7 \pm 0.9$ | 64.0 |
| SHOT-IM | $61.6 \pm 0.3$ | $\mathbf{82.1 \pm 0.3}$ | $\mathbf{62.5 \pm 0.3}$ | $32.8 \pm 0.4$ | 59.8 |
| SHOT | $61.8 \pm 0.3$ | $\mathbf{82.3 \pm 0.2}$ | $\mathbf{62.8 \pm 0.2}$ | $32.7 \pm 0.4$ | 59.9 |
| PL | $67.0 \pm 0.6$ | $72.9 \pm 1.0$ | $56.3 \pm 2.5$ | $35.4 \pm 1.7$ | 57.9 |
| PL-C | $71.8 \pm 1.3$ | $78.9 \pm 0.4$ | $61.7 \pm 0.3$ | $\mathbf{43.1 \pm 0.9}$ | 63.9 |
| Tent-Full | $72.3 \pm 0.3$ | $\mathbf{83.9 \pm 0.3}$ | $\mathbf{62.7 \pm 0.2}$ | $36.9 \pm 0.3$ | 64.0 |
| BN-Norm | $70.4 \pm 1.0$ | $\mathbf{82.7 \pm 0.1}$ | $\mathbf{62.0 \pm 0.1}$ | $36.4 \pm 0.2$ | 62.9 |
| Tent-C | $71.3 \pm 1.5$ | $74.6 \pm 1.9$ | $60.5 \pm 0.4$ | $40.9 \pm 0.5$ | 61.8 |
| Tent-BN | $64.7 \pm 0.7$ | $\mathbf{81.1 \pm 0.2}$ | $\mathbf{62.5 \pm 0.3}$ | $36.4 \pm 0.9$ | 61.2 |
| T3A (Ours) | $\mathbf{74.5 \pm 0.9}$ | $\mathbf{81.4 \pm 0.2}$ | $\mathbf{63.2 \pm 0.4}$ | $39.5 \pm 0.3$ | $64.6^*$ |
| resnet50 w/ BN | $74.3 \pm 0.5$ | $84.1 \pm 0.1$ | $66.9 \pm 0.2$ | $45.8 \pm 1.8$ | 67.8 |
| SHOT-IM | $61.5 \pm 1.7$ | $\mathbf{84.6 \pm 0.3}$ | $\mathbf{68.0 \pm 0.0}$ | $33.8 \pm 0.3$ | 62.0 |
| SHOT | $61.6 \pm 1.8$ | $\mathbf{84.8 \pm 0.5}$ | $\mathbf{68.0 \pm 0.0}$ | $34.6 \pm 0.3$ | 62.3 |
| PL | $63.4 \pm 1.8$ | $80.1 \pm 3.5$ | $61.3 \pm 1.5$ | $36.8 \pm 4.4$ | 60.4 |
| PL-C | $73.3 \pm 0.8$ | $\mathbf{84.7 \pm 0.3}$ | $66.4 \pm 0.3$ | $\mathbf{47.0 \pm 1.7}$ | 67.9 |
| Tent-Full | $\mathbf{75.4 \pm 0.6}$ | $\mathbf{87.0 \pm 0.2}$ | $66.9 \pm 0.2$ | $42.6 \pm 0.8$ | 68.0 |
| BN-Norm | $71.3 \pm 0.4$ | $\mathbf{85.8 \pm 0.1}$ | $66.4 \pm 0.1$ | $42.3 \pm 0.4$ | 66.5 |
| Tent-C | $72.4 \pm 1.5$ | $\mathbf{84.4 \pm 0.1}$ | $66.2 \pm 0.2$ | $42.4 \pm 3.1$ | 66.4 |
| Tent-BN | $65.6 \pm 1.4$ | $\mathbf{84.9 \pm 0.0}$ | $\mathbf{67.7 \pm 0.2}$ | $42.7 \pm 0.5$ | 65.2 |
| T3A (Ours) | $\mathbf{76.0 \pm 0.3}$ | $\mathbf{85.1 \pm 0.2}$ | $\mathbf{68.2 \pm 0.1}$ | $44.6 \pm 0.9$ | $68.5^*$ |

algorithms for training-phase (the third block of Table 1), and different backbone networks (Table 2). Note that this property is important in practice because a better backbone network usually give significant performance gains. For example, Table 2 suggest a practitioner should try HViT, which outperforms ResNet50 by a large margin (7.8 points in average) if computational resources allow. T3A can boost HViT's performance by 1.0 points. Unlike existing studies that update the model with SGD during testing, the proposed method is optimization-free. Therefore, the computational overhead is negligible, and the behavior is unlikely to become unstable, making T3A especially suitable for online settings where the model needs to (adapt and) predict online as with typical DG.

One of the limitations of T3A is how to extend it beyond the classification problem. Since the proposed method creates a template online for each prediction class, it is not trivial to adapt it for continuous prediction. Note that Tent have the same problem, as prediction entropy is hard to compute in the regression case. It is a future task to apply this idea to a broader range of problem settings.

Another potential drawback of the proposed method is that the model can change at any time, making it difficult to thoroughly test its behavior in advance. This may raise ethical concerns in some sensitive applications, making it more difficult to sanitize the model and ensure it does not make unfair decisions. In such a situation, fully online adaptation might be difficult, and one may want to update the model offline. We encourage examination of each of these works on the frontier of test-time adaptation.

Although accuracy was greatly improved, there is massive room for improvement as the performance in the unknown domain is still significantly worse than performance in the known domain. From a probabilistic viewpoint, the templates of each class can be regarded as the statistics of the $P(\boldsymbol{Z}|Y)$, and our method adjusts it. As it is connected to the prediction $P(Y|\boldsymbol{Z}) = \frac{P(\boldsymbol{Z}|Y)P(Y)}{P(\boldsymbol{Z})}$, adjusting it can correlates the prediction. From this perspective, using the average templates might be too restrictive, and one can use higher-order statistics to improve performance. Alternatively, one can retain all reliable samples, approximating $P(\boldsymbol{Z}|Y)$ as empirical distribution. We hope that the findings of this paper will lead to a better test-time adaptation method and lead to the development of machine learning systems that work well in unknown environments.

## Acknowledgements

This work has been supported by JSPS Grant-in-Aid for Early-Career Scientists Number JP18K18101 and the Mohammed bin Salman Center for Future Science and Technology for Saudi-Japan Vision 2030 at The University of Tokyo (MbSC2030). Computational resource of AI Bridging Cloud Infrastructure (ABCI) provided by National Institute of Advanced Industrial Science and Technology (AIST) was used.

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
