# A  Details of Experimental Setup

## A.1  Code

Code is available at https://github.com/matsuolab/T3A.

## A.2  Total amount of compute

We run our experiments mainly on cloud V100x4 or A100x8 instances, depending on the memory usage of the backbone networks. We used approximately 1500 hours for the v100x4 instances and 400 hours for the a100x8 instances. Note that the computational time was mainly consumed to reproduce prior results (training base models with ERM and CORAL). Our method only has negligible computational overhead, as mentioned in Section 3.2.

## A.3  License of assets

**Datasets**  VLCS is a combination of 4 datasets: Caltech101 (unspecified), PASCAL VOC (Flicks terms of use), LabelME (unspecified) and SUN09 (unspecified). PACS is a combination of several sources: Caltech256 (unspecified), Sketchy (Apache-2.0 for the download script, dataset unspecified), TU-Berlin (CC BY 4.0) and Google Images (unspecified). OfficeHome (non-commercial research and educational purposes). TerraIncognita (CDLA-Permissive-1.0).

**Codes**  `DomainBed` (MIT License), `torch-vision` for ResNet18 and ResNet50 (Apache-2.0), the official repository of Big Transfer (Apache-2.0), `timm` for Vision Transformer and MLP-Mixer (Apache-2.0), and the official repository of Tent (MIT License).

## A.4  Hyperparameters

Following [52], we conduct a random search of 20 trials over a joint distribution of all hyperparameters. Namely, for both ERM and CORAL, learning rate is selected from $10^{\text{Uniform}(-5, -3.5)}$, batch size is selected from $2^{\text{Uniform}(3, 5.5)}$, dropout rate is selected from $[0, 0.1, 0.5]$, and weight decay is selected from $10^{\text{Uniform}(-6, -2)}$.

## A.5  Trial seeds

For the results of Table 1 and Figure 2, we used three seeds $\{0, 1, 2\}$ in `DomainNet` library, which is the same with the original paper. For the results of Table 2 we used three seeds $\{10, 11, 12\}$ as well.

# B  Full Results

This section contains the full results of the entire experiment—for example, Appendix B.1 shows the full results of Table 1 per dataset and domain. Please refer to the main text for the detailed notation of each chart.

## B.1  Full Results for Table 1

| Algorithm | C | L | S | V | Avg |
|---|---|---|---|---|---|
| ERM | 97.7 ± 0.4 | 64.3 ± 0.9 | 73.4 ± 0.5 | 74.6 ± 1.3 | 77.5 |
| IRM | 98.6 ± 0.1 | 64.9 ± 0.9 | 73.4 ± 0.6 | 77.3 ± 0.9 | 78.5 |
| GroupDRO | 97.3 ± 0.3 | 63.4 ± 0.9 | 69.5 ± 0.8 | 76.7 ± 0.7 | 76.7 |
| Mixup | 98.3 ± 0.6 | 64.8 ± 1.0 | 72.1 ± 0.5 | 74.3 ± 0.8 | 77.4 |
| MLDG | 97.4 ± 0.2 | 65.2 ± 0.7 | 71.0 ± 1.4 | 75.3 ± 1.0 | 77.2 |
| CORAL | 98.3 ± 0.1 | 66.1 ± 1.2 | 73.4 ± 0.3 | 77.5 ± 1.2 | 78.8 |
| MMD | 97.7 ± 0.1 | 64.0 ± 1.1 | 72.8 ± 0.2 | 75.3 ± 3.3 | 77.5 |
| DANN | 99.0 ± 0.3 | 65.1 ± 1.4 | 73.1 ± 0.3 | 77.2 ± 0.6 | 78.6 |
| CDANN | 97.1 ± 0.3 | 65.1 ± 1.2 | 70.7 ± 0.8 | 77.1 ± 1.5 | 77.5 |
| MTL | 97.8 ± 0.4 | 64.3 ± 0.3 | 71.5 ± 0.7 | 75.3 ± 1.7 | 77.2 |
| SagNet | 97.9 ± 0.4 | 64.5 ± 0.5 | 71.4 ± 1.3 | 77.5 ± 0.5 | 77.8 |
| ARM | 98.7 ± 0.2 | 63.6 ± 0.7 | 71.3 ± 1.2 | 76.7 ± 0.6 | 77.6 |
| VREx | 98.4 ± 0.3 | 64.4 ± 1.4 | 74.1 ± 0.4 | 76.2 ± 1.3 | 78.3 |
| RSC | 97.9 ± 0.1 | 62.5 ± 0.7 | 72.3 ± 1.2 | 75.6 ± 0.8 | 77.1 |
| ERM** | 97.7 ± 0.1 | 64.1 ± 0.8 | 72.5 ± 0.9 | 76.7 ± 0.5 | 77.7 |
| +T3A | **99.1 ± 0.4** | **67.5 ± 0.6** | **76.8 ± 1.2** | 76.6 ± 0.3 | **80.0** |
| +Tent-BN | 83.3 ± 2.0 | 60.7 ± 0.2 | 61.7 ± 1.1 | 67.2 ± 0.8 | 68.2 |
| +Tent-C | **97.8 ± 0.2** | **64.3 ± 1.0** | 68.3 ± 0.8 | **77.5 ± 0.2** | 77.0 |
| CORAL** | 97.6 ± 0.2 | 65.1 ± 0.8 | 73.7 ± 1.0 | 78.0 ± 0.2 | 78.6 |
| +T3A | **98.9 ± 0.2** | **67.1 ± 1.0** | **75.2 ± 1.3** | 76.9 ± 0.6 | **79.5** |
| +Tent-BN | 87.1 ± 0.3 | 60.4 ± 1.7 | 67.8 ± 0.7 | 70.3 ± 0.6 | 71.4 |
| +Tent-C | 97.4 ± 0.2 | 64.7 ± 1.0 | 72.1 ± 0.7 | **78.3 ± 0.2** | 78.1 |

Table 4: Full results for Table 1 on VLCS.

| Algorithm | A | C | P | S | Avg |
|---|---|---|---|---|---|
| ERM | 84.7 ± 0.4 | 80.8 ± 0.6 | 97.2 ± 0.3 | 79.3 ± 1.0 | 85.5 |
| IRM | 84.8 ± 1.3 | 76.4 ± 1.1 | 96.7 ± 0.6 | 76.1 ± 1.0 | 83.5 |
| GroupDRO | 83.5 ± 0.9 | 79.1 ± 0.6 | 96.7 ± 0.3 | 78.3 ± 2.0 | 84.4 |
| Mixup | 86.1 ± 0.5 | 78.9 ± 0.8 | 97.6 ± 0.1 | 75.8 ± 1.8 | 84.6 |
| MLDG | 85.5 ± 1.4 | 80.1 ± 1.7 | 97.4 ± 0.3 | 76.6 ± 1.1 | 84.9 |
| CORAL | 88.3 ± 0.2 | 80.0 ± 0.5 | 97.5 ± 0.3 | 78.8 ± 1.3 | 86.2 |
| MMD | 86.1 ± 1.4 | 79.4 ± 0.9 | 96.6 ± 0.2 | 76.5 ± 0.5 | 84.6 |
| DANN | 86.4 ± 0.8 | 77.4 ± 0.8 | 97.3 ± 0.4 | 73.5 ± 2.3 | 83.6 |
| CDANN | 84.6 ± 1.8 | 75.5 ± 0.9 | 96.8 ± 0.3 | 73.5 ± 0.6 | 82.6 |
| MTL | 87.5 ± 0.8 | 77.1 ± 0.5 | 96.4 ± 0.8 | 77.3 ± 1.8 | 84.6 |
| SagNet | 87.4 ± 1.0 | 80.7 ± 0.6 | 97.1 ± 0.1 | 80.0 ± 0.4 | 86.3 |
| ARM | 86.8 ± 0.6 | 76.8 ± 0.5 | 97.4 ± 0.3 | 79.3 ± 1.2 | 85.1 |
| VREx | 86.0 ± 1.6 | 79.1 ± 0.6 | 96.9 ± 0.5 | 77.7 ± 1.7 | 84.9 |
| RSC | 85.4 ± 0.8 | 79.7 ± 1.8 | 97.6 ± 0.3 | 78.2 ± 1.2 | 85.2 |
| ERM** | 85.0 ± 0.8 | 77.9 ± 2.0 | 97.1 ± 0.5 | 74.6 ± 1.6 | 83.6 |
| +T3A | **86.3 ± 0.9** | **80.3 ± 0.9** | **97.5 ± 0.4** | **77.1 ± 1.4** | **85.3** |
| +Tent-BN | **86.3 ± 0.1** | **80.0 ± 0.9** | 96.9 ± 0.3 | **76.1 ± 1.0** | **84.8** |
| +Tent-C | 84.9 ± 0.9 | **78.2 ± 2.2** | 97.1 ± 0.5 | 68.9 ± 3.5 | 82.3 |
| CORAL** | 86.6 ± 0.5 | 77.0 ± 1.0 | 97.6 ± 0.1 | 75.7 ± 1.3 | 84.2 |
| +T3A | **87.7 ± 0.6** | **78.9 ± 0.6** | **97.8 ± 0.1** | **78.1 ± 0.8** | **85.6** |
| +Tent-BN | **87.9 ± 0.3** | **78.5 ± 0.7** | 97.5 ± 0.1 | **78.2 ± 0.3** | **85.6** |
| +Tent-C | **87.3 ± 1.0** | 76.7 ± 1.2 | 97.6 ± 0.1 | 73.3 ± 2.6 | 83.7 |

Table 5: Full results for Table 1 on PACS.

| Algorithm | A | C | P | R | Avg |
|---|---|---|---|---|---|
| ERM | $61.3 \pm 0.7$ | $52.4 \pm 0.3$ | $75.8 \pm 0.1$ | $76.6 \pm 0.3$ | 66.5 |
| IRM | $58.9 \pm 2.3$ | $52.2 \pm 1.6$ | $72.1 \pm 2.9$ | $74.0 \pm 2.5$ | 64.3 |
| GroupDRO | $60.4 \pm 0.7$ | $52.7 \pm 1.0$ | $75.0 \pm 0.7$ | $76.0 \pm 0.7$ | 66.0 |
| Mixup | $62.4 \pm 0.8$ | $54.8 \pm 0.6$ | $76.9 \pm 0.3$ | $78.3 \pm 0.2$ | 68.1 |
| MLDG | $61.5 \pm 0.9$ | $53.2 \pm 0.6$ | $75.0 \pm 1.2$ | $77.5 \pm 0.4$ | 66.8 |
| CORAL | $65.3 \pm 0.4$ | $54.4 \pm 0.5$ | $76.5 \pm 0.1$ | $78.4 \pm 0.5$ | 68.7 |
| MMD | $60.4 \pm 0.2$ | $53.3 \pm 0.3$ | $74.3 \pm 0.1$ | $77.4 \pm 0.6$ | 66.3 |
| DANN | $59.9 \pm 1.3$ | $53.0 \pm 0.3$ | $73.6 \pm 0.7$ | $76.9 \pm 0.5$ | 65.9 |
| CDANN | $61.5 \pm 1.4$ | $50.4 \pm 2.4$ | $74.4 \pm 0.9$ | $76.6 \pm 0.8$ | 65.8 |
| MTL | $61.5 \pm 0.7$ | $52.4 \pm 0.6$ | $74.9 \pm 0.4$ | $76.8 \pm 0.4$ | 66.4 |
| SagNet | $63.4 \pm 0.2$ | $54.8 \pm 0.4$ | $75.8 \pm 0.4$ | $78.3 \pm 0.3$ | 68.1 |
| ARM | $58.9 \pm 0.8$ | $51.0 \pm 0.5$ | $74.1 \pm 0.1$ | $75.2 \pm 0.3$ | 64.8 |
| VREx | $60.7 \pm 0.9$ | $53.0 \pm 0.9$ | $75.3 \pm 0.1$ | $76.6 \pm 0.5$ | 66.4 |
| RSC | $60.7 \pm 1.4$ | $51.4 \pm 0.3$ | $74.8 \pm 1.1$ | $75.1 \pm 1.3$ | 65.5 |
| ERM** | $59.8 \pm 0.3$ | $53.9 \pm 0.5$ | $75.1 \pm 0.2$ | $76.8 \pm 0.5$ | 66.4 |
| +T3A | $\mathbf{61.6 \pm 0.1}$ | $\mathbf{56.0 \pm 0.3}$ | $\mathbf{77.3 \pm 0.2}$ | $\mathbf{78.2 \pm 0.3}$ | **68.3** |
| +Tent-BN | $\mathbf{62.1 \pm 0.4}$ | $\mathbf{55.1 \pm 0.5}$ | $74.6 \pm 0.0$ | $76.2 \pm 0.6$ | **67.0** |
| +Tent-C | $59.2 \pm 0.2$ | $51.6 \pm 0.8$ | $\mathbf{75.2 \pm 0.3}$ | $76.7 \pm 0.3$ | 65.7 |
| CORAL** | $64.3 \pm 0.4$ | $54.3 \pm 0.2$ | $76.5 \pm 0.3$ | $78.0 \pm 0.4$ | 68.3 |
| +T3A | $64.2 \pm 0.3$ | $\mathbf{56.1 \pm 0.2}$ | $\mathbf{78.1 \pm 0.1}$ | $\mathbf{78.6 \pm 0.3}$ | **69.2** |
| +Tent-BN | $\mathbf{65.6 \pm 0.5}$ | $\mathbf{56.5 \pm 0.3}$ | $76.3 \pm 0.1$ | $\mathbf{78.4 \pm 0.3}$ | **69.2** |
| +Tent-C | $64.2 \pm 0.2$ | $54.0 \pm 0.2$ | $76.4 \pm 0.3$ | $\mathbf{78.1 \pm 0.4}$ | 68.2 |

Table 6: Full results for Table 1 on OfficeHome.

| Algorithm | L100 | L38 | L43 | L46 | Avg |
|---|---|---|---|---|---|
| ERM | $49.8 \pm 4.4$ | $42.1 \pm 1.4$ | $56.9 \pm 1.8$ | $35.7 \pm 3.9$ | 46.1 |
| IRM | $54.6 \pm 1.3$ | $39.8 \pm 1.9$ | $56.2 \pm 1.8$ | $39.6 \pm 0.8$ | 47.6 |
| GroupDRO | $41.2 \pm 0.7$ | $38.6 \pm 2.1$ | $56.7 \pm 0.9$ | $36.4 \pm 2.1$ | 43.2 |
| Mixup | $59.6 \pm 2.0$ | $42.2 \pm 1.4$ | $55.9 \pm 0.8$ | $33.9 \pm 1.4$ | 47.9 |
| MLDG | $54.2 \pm 3.0$ | $44.3 \pm 1.1$ | $55.6 \pm 0.3$ | $36.9 \pm 2.2$ | 47.7 |
| CORAL | $51.6 \pm 2.4$ | $42.2 \pm 1.0$ | $57.0 \pm 1.0$ | $39.8 \pm 2.9$ | 47.6 |
| MMD | $41.9 \pm 3.0$ | $34.8 \pm 1.0$ | $57.0 \pm 1.9$ | $35.2 \pm 1.8$ | 42.2 |
| DANN | $51.1 \pm 3.5$ | $40.6 \pm 0.6$ | $57.4 \pm 0.5$ | $37.7 \pm 1.8$ | 46.7 |
| CDANN | $47.0 \pm 1.9$ | $41.3 \pm 4.8$ | $54.9 \pm 1.7$ | $39.8 \pm 2.3$ | 45.8 |
| MTL | $49.3 \pm 1.2$ | $39.6 \pm 6.3$ | $55.6 \pm 1.1$ | $37.8 \pm 0.8$ | 45.6 |
| SagNet | $53.0 \pm 2.9$ | $43.0 \pm 2.5$ | $57.9 \pm 0.6$ | $40.4 \pm 1.3$ | 48.6 |
| ARM | $49.3 \pm 0.7$ | $38.3 \pm 2.4$ | $55.8 \pm 0.8$ | $38.7 \pm 1.3$ | 45.5 |
| VREx | $48.2 \pm 4.3$ | $41.7 \pm 1.3$ | $56.8 \pm 0.8$ | $38.7 \pm 3.1$ | 46.4 |
| RSC | $50.2 \pm 2.2$ | $39.2 \pm 1.4$ | $56.3 \pm 1.4$ | $40.8 \pm 0.6$ | 46.6 |
| ERM** | $50.4 \pm 2.4$ | $45.1 \pm 0.7$ | $52.3 \pm 1.3$ | $38.1 \pm 0.5$ | 46.5 |
| +T3A | $48.6 \pm 2.1$ | $\mathbf{45.7 \pm 1.1}$ | $52.1 \pm 1.5$ | $\mathbf{41.5 \pm 1.5}$ | **47.0** |
| +Tent-BN | $\mathbf{53.1 \pm 1.3}$ | $42.6 \pm 0.9$ | $46.6 \pm 0.5$ | $36.5 \pm 0.5$ | 44.7 |
| +Tent-C | $\mathbf{51.2 \pm 1.7}$ | $44.9 \pm 0.5$ | $50.4 \pm 0.7$ | $35.5 \pm 0.9$ | 45.5 |
| CORAL** | $54.0 \pm 3.4$ | $43.3 \pm 1.3$ | $55.5 \pm 1.4$ | $39.4 \pm 0.3$ | 48.1 |
| +T3A | $49.1 \pm 1.2$ | $\mathbf{44.3 \pm 1.1}$ | $54.6 \pm 0.7$ | $\mathbf{41.1 \pm 0.7}$ | 47.3 |
| +Tent-BN | $51.0 \pm 0.8$ | $\mathbf{44.2 \pm 0.9}$ | $52.2 \pm 1.1$ | $38.7 \pm 0.6$ | 46.5 |
| +Tent-C | $\mathbf{55.0 \pm 3.7}$ | $42.1 \pm 0.6$ | $55.5 \pm 1.6$ | $38.7 \pm 0.3$ | 47.8 |

Table 7: Full results for Table 1 on TerraIncognita.

## B.2 Full Results for Table 2

Table 8: Full results for Table 2 of resnet18.

(a) VLCS

|  | C | L | S | V | Avg |
|---|---|---|---|---|---|
| ERM | 95.8 ± 0.4 | 60.4 ± 1.1 | 66.4 ± 0.8 | 70.2 ± 1.5 | 73.2 |
| +T3C | **99.2 ± 0.3** | **64.5 ± 1.2** | **69.4 ± 1.0** | **73.0 ± 1.9** | 76.5 |
| +Tent-BN | 82.6 ± 0.6 | 50.8 ± 1.2 | 54.5 ± 1.7 | 57.3 ± 1.5 | 61.3 |
| +Tent-C | **96.1 ± 0.7** | 57.4 ± 2.1 | 65.1 ± 1.1 | 67.4 ± 2.1 | 71.5 |

(b) PACS

|  | A | C | P | S | Avg |
|---|---|---|---|---|---|
| ERM | 78.7 ± 1.3 | 74.3 ± 0.6 | 92.4 ± 0.2 | 75.6 ± 0.8 | 80.3 |
| +T3C | **80.4 ± 0.7** | 75.2 ± 0.4 | **94.7 ± 0.5** | **76.5 ± 0.2** | 81.7 |
| +Tent-BN | 78.8 ± 0.8 | 76.3 ± 0.4 | 92.8 ± 0.3 | 73.9 ± 0.2 | 80.5 |
| +Tent-C | 78.1 ± 1.8 | 74.3 ± 0.6 | 92.5 ± 0.3 | 75.0 ± 1.0 | 80.0 |

(c) OfficeHome

|  | A | C | P | R | Avg |
|---|---|---|---|---|---|
| ERM | 46.1 ± 0.4 | 45.8 ± 0.3 | 65.0 ± 0.3 | 66.0 ± 0.3 | 55.7 |
| +T3C | **47.0 ± 0.2** | **46.8 ± 1.0** | **68.0 ± 0.2** | 66.1 ± 0.4 | 57.0 |
| +Tent-BN | **46.5 ± 0.1** | 44.6 ± 0.4 | 62.1 ± 0.4 | 62.5 ± 0.2 | 53.9 |
| +Tent-C | 46.0 ± 0.4 | 45.5 ± 0.4 | 64.9 ± 0.3 | **66.1 ± 0.4** | 55.6 |

(d) TerraIncognita

|  | L100 | L38 | L43 | L46 | Avg |
|---|---|---|---|---|---|
| ERM | 41.6 ± 3.1 | 36.0 ± 3.0 | 51.0 ± 0.2 | 33.9 ± 1.0 | 40.7 |
| +T3C | 48.6 ± 2.5 | **41.0 ± 1.9** | 44.7 ± 0.7 | 31.8 ± 0.9 | 41.6 |
| +Tent-BN | 49.6 ± 0.2 | 36.6 ± 2.0 | 39.8 ± 1.4 | 33.1 ± 1.7 | 39.8 |
| +Tent-C | **43.9 ± 3.3** | 34.5 ± 3.3 | 48.8 ± 1.2 | 32.7 ± 1.7 | 40.0 |

Table 9: Full results for Table 2 of resnet50.

(a) VLCS

|  | C | L | S | V | Avg |
|---|---|---|---|---|---|
| ERM | 97.6 ± 0.1 | 63.4 ± 0.5 | 69.3 ± 1.6 | 72.0 ± 1.5 | 75.5 |
| +T3C | **99.1 ± 0.1** | **67.2 ± 0.8** | **73.7 ± 1.7** | **73.2 ± 1.5** | 78.3 |
| +Tent-BN | 87.6 ± 0.9 | 61.7 ± 1.2 | 65.8 ± 1.4 | 64.6 ± 1.7 | 69.9 |
| +Tent-C | 94.1 ± 3.2 | **64.8 ± 0.5** | 67.0 ± 2.7 | 64.1 ± 2.2 | 72.5 |

(b) PACS

|  | A | C | P | S | Avg |
|---|---|---|---|---|---|
| ERM | 84.8 ± 0.5 | 78.5 ± 0.6 | 96.1 ± 0.2 | 76.2 ± 1.1 | 83.9 |
| +T3C | **86.0 ± 0.6** | **80.3 ± 0.9** | **96.4 ± 0.2** | 75.2 ± 1.5 | 84.5 |
| +Tent-BN | 84.9 ± 0.4 | 79.8 ± 0.6 | 96.0 ± 0.4 | 75.7 ± 0.7 | 84.1 |
| +Tent-C | 73.5 ± 6.3 | 73.7 ± 4.0 | 96.0 ± 0.4 | 75.1 ± 1.6 | 79.6 |

(c) OfficeHome

|  | A | C | P | R | Avg |
|---|---|---|---|---|---|
| ERM | 58.5 ± 0.4 | 50.7 ± 1.0 | 73.3 ± 0.5 | 75.0 ± 0.2 | 64.4 |
| +T3C | **60.2 ± 0.3** | **53.4 ± 0.9** | **76.1 ± 0.4** | **76.2 ± 0.1** | 66.5 |
| +Tent-BN | **59.5 ± 0.2** | 52.5 ± 1.2 | 72.3 ± 0.1 | 73.9 ± 0.2 | 64.6 |
| +Tent-C | 57.8 ± 0.6 | 49.1 ± 2.1 | 73.1 ± 0.6 | 74.9 ± 0.3 | 63.7 |

(d) TerraIncognita

|  | L100 | L38 | L43 | L46 | Avg |
|---|---|---|---|---|---|
| ERM | 53.4 ± 4.1 | 39.0 ± 1.8 | 52.0 ± 1.0 | 37.1 ± 2.8 | 45.4 |
| +T3C | 52.7 ± 2.7 | **40.6 ± 2.2** | 51.9 ± 1.2 | **38.5 ± 2.8** | 45.9 |
| +Tent-BN | **54.1 ± 2.7** | 38.9 ± 1.9 | 44.6 ± 0.9 | 36.1 ± 1.6 | 43.4 |
| +Tent-C | 38.1 ± 14.2 | 37.7 ± 2.4 | 51.8 ± 1.3 | 35.3 ± 2.9 | 40.7 |

Table 10: Full results for Table 2 of BiT-M-R50x3.

(a) VLCS

|  | C | L | S | V | Avg |
|---|---|---|---|---|---|
| ERM | 93.2 ± 0.4 | 66.1 ± 0.3 | 74.0 ± 0.4 | 73.3 ± 1.2 | 76.7 |
| +T3C | **97.5 ± 0.4** | **67.5 ± 0.9** | **77.2 ± 0.5** | **76.5 ± 1.8** | 79.7 |
| +Tent-BN | 89.5 ± 0.8 | 66.1 ± 0.8 | 69.2 ± 0.8 | 69.7 ± 1.2 | 73.6 |
| +Tent-C | 93.4 ± 0.7 | 60.9 ± 4.5 | 60.0 ± 6.7 | 72.2 ± 1.6 | 71.6 |

(b) PACS

|  | A | C | P | S | Avg |
|---|---|---|---|---|---|
| ERM | 82.0 ± 1.7 | 82.3 ± 0.9 | 95.1 ± 0.4 | 78.1 ± 2.1 | 84.4 |
| +T3C | **83.1 ± 2.0** | **83.8 ± 0.2** | **96.4 ± 0.2** | 78.5 ± 1.4 | 85.4 |
| +Tent-BN | 83.0 ± 1.4 | 83.0 ± 0.4 | 96.3 ± 0.4 | 79.0 ± 1.2 | 85.3 |
| +Tent-C | **83.3 ± 1.8** | 81.4 ± 1.2 | 95.2 ± 0.4 | 77.8 ± 2.1 | 84.4 |

(c) OfficeHome

|  | A | C | P | R | Avg |
|---|---|---|---|---|---|
| ERM | 64.3 ± 1.2 | 56.0 ± 0.9 | 78.1 ± 0.8 | 78.3 ± 0.7 | 69.2 |
| +T3C | 65.8 ± 1.2 | 59.5 ± 1.4 | 80.9 ± 1.0 | 80.5 ± 0.8 | 71.7 |
| +Tent-BN | 66.6 ± 0.6 | 58.1 ± 1.4 | 79.3 ± 1.1 | 79.0 ± 0.8 | 70.8 |
| +Tent-C | 63.8 ± 1.4 | 55.3 ± 1.0 | 78.1 ± 0.8 | **78.5 ± 0.8** | 68.9 |

(d) TerraIncognita

|  | L100 | L38 | L43 | L46 | Avg |
|---|---|---|---|---|---|
| ERM | 60.6 ± 1.4 | 43.5 ± 2.9 | 59.0 ± 0.8 | 46.6 ± 2.9 | 52.5 |
| +T3C | **60.8 ± 0.5** | **44.0 ± 1.9** | 58.2 ± 0.9 | 45.9 ± 4.1 | 52.2 |
| +Tent-BN | 57.3 ± 0.5 | **43.9 ± 1.6** | 54.1 ± 0.7 | 46.6 ± 2.6 | 50.5 |
| +Tent-C | 55.7 ± 4.1 | 40.6 ± 3.7 | **59.2 ± 1.0** | 46.5 ± 2.7 | 50.5 |

## Table 11: Full results for Table 2 of BiT-M-R101x3.

### (a) VLCS

|  | C | L | S | V | Avg |
|---|---|---|---|---|---|
| ERM | 93.4 ± 1.4 | 62.7 ± 1.4 | 71.3 ± 0.7 | 72.6 ± 1.3 | 75.0 |
| +T3C | **97.6 ± 0.7** | **65.9 ± 1.5** | **75.6 ± 0.8** | **75.4 ± 1.4** | 78.6 |
| +Tent-BN | 89.0 ± 2.2 | 63.1 ± 0.4 | 69.3 ± 0.6 | 68.1 ± 0.9 | 72.4 |
| +Tent-C | **95.5 ± 1.4** | 62.5 ± 1.5 | 65.3 ± 1.9 | 71.7 ± 0.5 | 73.8 |

### (b) PACS

|  | A | C | P | S | Avg |
|---|---|---|---|---|---|
| ERM | 81.1 ± 0.1 | 83.0 ± 0.8 | 94.0 ± 0.5 | 77.7 ± 1.5 | 84.0 |
| +T3C | 81.6 ± 0.1 | **83.9 ± 0.8** | **95.2 ± 0.5** | **80.7 ± 0.9** | 85.4 |
| +Tent-BN | **82.7 ± 0.7** | **84.1 ± 0.4** | 94.9 ± 0.4 | **81.2 ± 0.8** | 85.7 |
| +Tent-C | **81.5 ± 0.4** | 79.3 ± 1.9 | 88.9 ± 4.1 | 74.4 ± 3.6 | 81.0 |

### (c) OfficeHome

|  | A | C | P | R | Avg |
|---|---|---|---|---|---|
| ERM | 59.7 ± 1.2 | 55.4 ± 1.1 | 78.7 ± 0.2 | 76.9 ± 0.2 | 67.7 |
| +T3C | **61.3 ± 1.2** | **58.2 ± 1.0** | **81.2 ± 0.3** | **78.9 ± 0.2** | 69.9 |
| +Tent-BN | **62.1 ± 1.2** | 57.4 ± 1.1 | **79.9 ± 0.4** | **77.6 ± 0.0** | 69.2 |
| +Tent-C | 59.6 ± 1.3 | 54.7 ± 1.1 | 78.7 ± 0.2 | **77.0 ± 0.3** | 67.5 |

### (d) TerraIncognita

|  | L100 | L38 | L43 | L46 | Avg |
|---|---|---|---|---|---|
| ERM | 54.5 ± 4.2 | 39.9 ± 2.6 | 56.5 ± 0.7 | 40.5 ± 0.5 | 47.8 |
| +T3C | **55.2 ± 3.8** | **41.3 ± 2.4** | 55.8 ± 1.0 | 40.1 ± 1.5 | 48.1 |
| +Tent-BN | 54.1 ± 2.3 | **41.2 ± 1.6** | 54.6 ± 0.4 | **41.4 ± 0.3** | 47.8 |
| +Tent-C | 49.6 ± 3.9 | 38.2 ± 3.2 | 55.9 ± 0.7 | 40.1 ± 0.6 | 46.0 |

## Table 12: Full results for Table 2 of BiT-M-R152x2.

### (a) VLCS

|  | C | L | S | V | Avg |
|---|---|---|---|---|---|
| ERM | 95.6 ± 1.0 | 66.0 ± 1.0 | 73.3 ± 1.4 | 71.8 ± 0.7 | 76.7 |
| +T3C | **98.9 ± 0.2** | **68.2 ± 0.8** | 76.4 ± 2.2 | 72.8 ± 0.6 | 79.1 |
| +Tent-BN | 92.6 ± 1.2 | 64.6 ± 0.1 | 70.3 ± 1.8 | 68.0 ± 0.9 | 73.9 |
| +Tent-C | 93.3 ± 0.2 | 64.0 ± 0.5 | **74.1 ± 1.4** | 70.3 ± 2.3 | 75.4 |

### (b) PACS

|  | A | C | P | S | Avg |
|---|---|---|---|---|---|
| ERM | 80.3 ± 1.4 | 82.2 ± 1.1 | 95.8 ± 0.6 | 82.6 ± 0.2 | 85.2 |
| +T3C | **81.9 ± 1.3** | **83.7 ± 1.1** | **96.3 ± 0.5** | **83.6 ± 0.4** | 86.4 |
| +Tent-BN | **81.6 ± 1.4** | 82.9 ± 0.8 | 95.6 ± 0.5 | 82.8 ± 0.4 | 85.7 |
| +Tent-C | 77.9 ± 1.6 | 68.2 ± 9.7 | 95.1 ± 0.7 | **83.2 ± 0.6** | 81.1 |

### (c) OfficeHome

|  | A | C | P | R | Avg |
|---|---|---|---|---|---|
| ERM | 63.2 ± 0.7 | 60.2 ± 1.0 | 79.7 ± 0.6 | 82.2 ± 0.3 | 71.3 |
| +T3C | **64.4 ± 0.4** | **62.8 ± 1.4** | **82.0 ± 0.7** | **83.5 ± 0.2** | 73.2 |
| +Tent-BN | **64.7 ± 0.4** | 62.1 ± 0.9 | 80.4 ± 0.5 | 82.3 ± 0.2 | 72.3 |
| +Tent-C | 63.1 ± 0.6 | 59.4 ± 1.2 | 79.7 ± 0.7 | 82.1 ± 0.4 | 71.1 |

### (d) TerraIncognita

|  | L100 | L38 | L43 | L46 | Avg |
|---|---|---|---|---|---|
| ERM | 57.3 ± 1.4 | 49.3 ± 1.5 | 58.8 ± 0.5 | 40.1 ± 2.5 | 51.4 |
| +T3C | 57.3 ± 0.3 | 48.3 ± 1.7 | 57.7 ± 1.2 | 40.2 ± 2.0 | 50.9 |
| +Tent-BN | 55.1 ± 0.7 | 47.1 ± 1.9 | 55.4 ± 0.4 | 41.2 ± 1.8 | 49.7 |
| +Tent-C | 31.6 ± 9.3 | 48.5 ± 1.3 | 58.2 ± 0.1 | **40.7 ± 3.3** | 44.8 |

## Table 13: Full results for Table 2 of ViT-B16.

### (a) VLCS

|  | C | L | S | V | Avg |
|---|---|---|---|---|---|
| ERM | 97.1 ± 0.5 | 65.4 ± 1.2 | 76.6 ± 0.7 | 77.8 ± 0.5 | 79.2 |
| +T3C | **97.7 ± 0.4** | **66.8 ± 0.8** | **78.3 ± 0.8** | 78.0 ± 0.7 | 80.2 |
| +Tent-BN | 93.1 ± 0.9 | **67.3 ± 0.6** | 72.6 ± 0.8 | 75.9 ± 0.4 | 77.2 |
| +Tent-C | **98.1 ± 0.4** | 65.9 ± 0.9 | 76.3 ± 0.8 | **79.1 ± 0.4** | 79.9 |

### (b) PACS

|  | A | C | P | S | Avg |
|---|---|---|---|---|---|
| ERM | 89.9 ± 0.7 | 86.3 ± 0.4 | 99.2 ± 0.1 | 67.2 ± 0.1 | 85.7 |
| +T3C | 90.1 ± 0.7 | **86.7 ± 0.5** | **99.3 ± 0.1** | **67.9 ± 0.3** | 86.0 |
| +Tent-BN | 89.8 ± 0.6 | 86.4 ± 0.5 | **99.3 ± 0.1** | **69.2 ± 0.9** | 86.2 |
| +Tent-C | **90.8 ± 0.8** | **86.5 ± 0.4** | **99.3 ± 0.1** | 67.1 ± 0.1 | 85.9 |

### (c) OfficeHome

|  | A | C | P | R | Avg |
|---|---|---|---|---|---|
| ERM | 76.5 ± 0.6 | 62.7 ± 0.2 | 86.9 ± 0.1 | 87.6 ± 0.3 | 78.4 |
| +T3C | **77.3 ± 0.8** | 63.1 ± 0.1 | **87.3 ± 0.1** | **87.9 ± 0.3** | 78.9 |
| +Tent-BN | **77.1 ± 0.7** | **63.3 ± 0.1** | 87.0 ± 0.2 | 87.7 ± 0.2 | 78.8 |
| +Tent-C | 76.3 ± 0.6 | 62.1 ± 0.5 | **87.0 ± 0.1** | 87.7 ± 0.3 | 78.3 |

### (d) TerraIncognita

|  | L100 | L38 | L43 | L46 | Avg |
|---|---|---|---|---|---|
| ERM | 53.1 ± 1.9 | 26.0 ± 1.0 | 50.3 ± 0.6 | 37.9 ± 0.6 | 41.8 |
| +T3C | 52.6 ± 1.9 | **28.4 ± 0.7** | 49.7 ± 0.9 | **39.2 ± 0.8** | 42.5 |
| +Tent-BN | 48.0 ± 1.4 | **30.6 ± 0.1** | 47.6 ± 0.7 | 37.0 ± 0.7 | 40.8 |
| +Tent-C | 52.6 ± 2.1 | 26.1 ± 1.2 | 50.0 ± 1.2 | 37.4 ± 0.6 | 41.5 |

## Table 14: Full results for Table 2 of ViT-L16.

### (a) VLCS

|  | C | L | S | V | Avg |
|---|---|---|---|---|---|
| ERM | 97.7 ± 0.2 | 62.8 ± 1.9 | 74.1 ± 0.2 | 78.1 ± 1.1 | 78.2 |
| +T3C | **98.4 ± 0.1** | 63.5 ± 2.1 | **75.3 ± 0.2** | **78.8 ± 1.1** | 79.0 |
| +Tent-BN | 94.5 ± 0.3 | **65.4 ± 1.3** | 69.4 ± 0.2 | 76.0 ± 0.8 | 76.3 |
| +Tent-C | **98.9 ± 0.1** | 63.3 ± 2.1 | 73.9 ± 0.3 | **78.2 ± 1.5** | 78.6 |

### (b) PACS

|  | A | C | P | S | Avg |
|---|---|---|---|---|---|
| ERM | 88.8 ± 0.6 | 83.3 ± 0.5 | 98.6 ± 0.1 | 67.5 ± 2.8 | 84.6 |
| +T3C | **89.1 ± 0.6** | 84.6 ± 0.5 | 98.4 ± 0.2 | **69.7 ± 2.9** | 85.5 |
| +Tent-BN | **89.4 ± 0.6** | **85.3 ± 0.3** | 98.5 ± 0.1 | **70.3 ± 2.0** | 85.9 |
| +Tent-C | 88.8 ± 0.8 | 82.4 ± 0.4 | 98.3 ± 0.1 | 67.0 ± 3.1 | 84.1 |

### (c) OfficeHome

|  | A | C | P | R | Avg |
|---|---|---|---|---|---|
| ERM | 77.2 ± 1.2 | 61.6 ± 0.9 | 85.3 ± 1.0 | 87.9 ± 0.4 | 78.0 |
| +T3C | **77.7 ± 1.1** | **62.7 ± 1.0** | **86.2 ± 0.8** | **88.1 ± 0.5** | 78.7 |
| +Tent-BN | 77.5 ± 0.9 | 62.6 ± 0.8 | 85.7 ± 0.9 | 88.0 ± 0.4 | 78.5 |
| +Tent-C | 75.6 ± 0.3 | 61.3 ± 0.9 | 85.2 ± 0.9 | 87.9 ± 0.4 | 77.5 |

### (d) TerraIncognita

|  | L100 | L38 | L43 | L46 | Avg |
|---|---|---|---|---|---|
| ERM | 56.0 ± 1.3 | 38.9 ± 2.7 | 40.7 ± 8.0 | 35.2 ± 1.6 | 42.7 |
| +T3C | 56.5 ± 1.1 | **40.6 ± 2.8** | 48.3 ± 0.8 | **35.9 ± 2.5** | 45.3 |
| +Tent-BN | 49.8 ± 0.8 | 39.3 ± 2.4 | 46.4 ± 0.8 | 35.1 ± 2.1 | 42.6 |
| +Tent-C | **61.6 ± 0.4** | 38.5 ± 2.7 | **49.3 ± 0.5** | **35.5 ± 1.7** | 46.2 |

### Table 15: Full results for Table 2 of DeiT.

**(a) VLCS**

|  | C | L | S | V | Avg |
|---|---|---|---|---|---|
| ERM | $97.3 \pm 0.4$ | $64.0 \pm 0.6$ | $76.9 \pm 0.6$ | $78.9 \pm 0.8$ | 79.3 |
| +T3C | $\mathbf{98.6 \pm 0.2}$ | $\mathbf{66.1 \pm 0.6}$ | $\mathbf{79.8 \pm 0.5}$ | $\mathbf{80.8 \pm 0.4}$ | 81.3 |
| +Tent-BN | $91.5 \pm 0.5$ | $\mathbf{66.4 \pm 0.8}$ | $73.0 \pm 0.5$ | $76.5 \pm 0.6$ | 76.9 |
| +Tent-C | $\mathbf{99.4 \pm 0.1}$ | $63.7 \pm 1.4$ | $\mathbf{78.1 \pm 1.6}$ | $\mathbf{79.3 \pm 1.0}$ | 80.1 |

**(b) PACS**

|  | A | C | P | S | Avg |
|---|---|---|---|---|---|
| ERM | $92.8 \pm 0.5$ | $83.8 \pm 0.5$ | $98.5 \pm 0.1$ | $76.2 \pm 1.4$ | 87.8 |
| +T3C | $\mathbf{93.4 \pm 0.3}$ | $\mathbf{85.9 \pm 0.4}$ | $\mathbf{98.8 \pm 0.1}$ | $80.0 \pm 1.1$ | 89.5 |
| +Tent-BN | $92.5 \pm 0.5$ | $84.9 \pm 0.4$ | $99.0 \pm 0.1$ | $81.5 \pm 0.9$ | 89.5 |
| +Tent-C | $\mathbf{94.3 \pm 0.4}$ | $83.7 \pm 0.7$ | $\mathbf{98.7 \pm 0.1}$ | $77.1 \pm 2.1$ | 88.4 |

**(c) OfficeHome**

|  | A | C | P | R | Avg |
|---|---|---|---|---|---|
| ERM | $74.5 \pm 0.2$ | $63.3 \pm 0.6$ | $83.2 \pm 0.2$ | $85.5 \pm 0.2$ | 76.6 |
| +T3C | $\mathbf{76.1 \pm 0.2}$ | $\mathbf{65.6 \pm 0.6}$ | $\mathbf{85.0 \pm 0.2}$ | $\mathbf{86.5 \pm 0.1}$ | 78.3 |
| +Tent-BN | $\mathbf{74.8 \pm 0.4}$ | $64.3 \pm 0.5$ | $83.7 \pm 0.1$ | $85.7 \pm 0.3$ | 77.1 |
| +Tent-C | $73.5 \pm 0.4$ | $\mathbf{63.4 \pm 0.7}$ | $83.2 \pm 0.2$ | $85.6 \pm 0.3$ | 76.4 |

**(d) TerraIncognita**

|  | L100 | L38 | L43 | L46 | Avg |
|---|---|---|---|---|---|
| ERM | $57.5 \pm 2.9$ | $44.0 \pm 2.0$ | $56.0 \pm 0.4$ | $42.3 \pm 0.8$ | 50.0 |
| +T3C | $\mathbf{59.8 \pm 1.9}$ | $43.8 \pm 1.4$ | $54.8 \pm 0.4$ | $42.1 \pm 1.4$ | 50.1 |
| +Tent-BN | $52.0 \pm 2.1$ | $41.7 \pm 1.2$ | $52.3 \pm 0.8$ | $41.4 \pm 1.1$ | 46.9 |
| +Tent-C | $\mathbf{59.1 \pm 1.6}$ | $43.5 \pm 2.0$ | $55.8 \pm 0.4$ | $41.7 \pm 0.8$ | 50.0 |

### Table 16: Full results for Table 2 of HViT.

**(a) VLCS**

|  | C | L | S | V | Avg |
|---|---|---|---|---|---|
| ERM | $96.8 \pm 0.5$ | $64.1 \pm 0.9$ | $75.9 \pm 1.1$ | $80.0 \pm 1.2$ | 79.2 |
| +T3C | $\mathbf{97.6 \pm 0.5}$ | $\mathbf{66.8 \pm 1.1}$ | $\mathbf{78.4 \pm 1.6}$ | $\mathbf{81.4 \pm 1.2}$ | 81.0 |
| +Tent-BN | $92.2 \pm 0.1$ | $66.1 \pm 0.8$ | $73.0 \pm 1.2$ | $77.8 \pm 0.7$ | 77.3 |
| +Tent-C | $\mathbf{98.4 \pm 0.7}$ | $60.7 \pm 4.8$ | $75.3 \pm 0.9$ | $\mathbf{80.8 \pm 1.6}$ | 78.8 |

**(b) PACS**

|  | A | C | P | S | Avg |
|---|---|---|---|---|---|
| ERM | $89.5 \pm 1.4$ | $85.9 \pm 2.6$ | $98.0 \pm 0.5$ | $85.5 \pm 1.1$ | 89.7 |
| +T3C | $\mathbf{90.1 \pm 1.3}$ | $87.3 \pm 1.9$ | $98.4 \pm 0.3$ | $85.9 \pm 1.3$ | 90.4 |
| +Tent-BN | $89.8 \pm 1.2$ | $\mathbf{87.5 \pm 1.4}$ | $98.4 \pm 0.4$ | $\mathbf{86.2 \pm 0.8}$ | 90.5 |
| +Tent-C | $\mathbf{90.0 \pm 0.7}$ | $83.2 \pm 4.0$ | $98.4 \pm 0.3$ | $85.8 \pm 1.3$ | 89.3 |

**(c) OfficeHome**

|  | A | C | P | R | Avg |
|---|---|---|---|---|---|
| ERM | $77.3 \pm 1.2$ | $68.4 \pm 0.6$ | $87.0 \pm 0.2$ | $87.5 \pm 0.3$ | 80.0 |
| +T3C | $\mathbf{78.0 \pm 1.1}$ | $\mathbf{69.0 \pm 0.5}$ | $\mathbf{87.2 \pm 0.2}$ | $\mathbf{87.9 \pm 0.3}$ | 80.5 |
| +Tent-BN | $\mathbf{77.9 \pm 1.1}$ | $68.7 \pm 0.4$ | $87.1 \pm 0.1$ | $87.7 \pm 0.2$ | 80.3 |
| +Tent-C | $77.3 \pm 1.2$ | $68.1 \pm 0.5$ | $86.8 \pm 0.2$ | $87.5 \pm 0.2$ | 79.9 |

**(d) TerraIncognita**

|  | L100 | L38 | L43 | L46 | Avg |
|---|---|---|---|---|---|
| ERM | $62.3 \pm 2.2$ | $44.6 \pm 0.5$ | $56.6 \pm 0.6$ | $41.9 \pm 1.0$ | 51.4 |
| +T3C | $\mathbf{62.6 \pm 2.5}$ | $\mathbf{46.4 \pm 0.4}$ | $\mathbf{57.2 \pm 0.7}$ | $\mathbf{43.1 \pm 0.8}$ | 52.3 |
| +Tent-BN | $58.5 \pm 1.8$ | $45.3 \pm 0.5$ | $55.1 \pm 0.9$ | $42.7 \pm 0.4$ | 50.4 |
| +Tent-C | $62.0 \pm 2.1$ | $44.2 \pm 0.4$ | $55.2 \pm 1.1$ | $\mathbf{42.0 \pm 1.0}$ | 50.8 |

### Table 17: Full results for Table 2 of Mixer-L16.

**(a) VLCS**

|  | C | L | S | V | Avg |
|---|---|---|---|---|---|
| ERM | $98.8 \pm 0.2$ | $61.0 \pm 0.4$ | $72.5 \pm 0.9$ | $73.5 \pm 0.3$ | 76.4 |
| +T3C | $\mathbf{99.7 \pm 0.1}$ | $\mathbf{65.8 \pm 0.4}$ | $\mathbf{77.3 \pm 1.4}$ | $\mathbf{78.3 \pm 1.2}$ | 80.3 |
| +Tent-BN | $92.6 \pm 0.5$ | $59.9 \pm 1.9$ | $67.7 \pm 1.3$ | $71.1 \pm 1.0$ | 72.8 |
| +Tent-C | $\mathbf{99.2 \pm 0.3}$ | $57.2 \pm 3.0$ | $71.3 \pm 0.4$ | $\mathbf{73.7 \pm 1.7}$ | 75.3 |

**(b) PACS**

|  | A | C | P | S | Avg |
|---|---|---|---|---|---|
| ERM | $79.9 \pm 3.2$ | $80.3 \pm 0.8$ | $97.6 \pm 0.4$ | $67.5 \pm 0.3$ | 81.3 |
| +T3C | $\mathbf{81.8 \pm 3.1}$ | $\mathbf{82.3 \pm 0.5}$ | $\mathbf{98.7 \pm 0.3}$ | $\mathbf{69.3 \pm 0.7}$ | 83.0 |
| +Tent-BN | $\mathbf{81.7 \pm 3.2}$ | $81.6 \pm 0.6$ | $97.5 \pm 0.4$ | $67.5 \pm 0.7$ | 82.1 |
| +Tent-C | $80.5 \pm 2.9$ | $77.4 \pm 1.4$ | $\mathbf{97.7 \pm 0.4}$ | $65.7 \pm 1.3$ | 80.3 |

**(c) OfficeHome**

|  | A | C | P | R | Avg |
|---|---|---|---|---|---|
| ERM | $69.9 \pm 2.4$ | $51.3 \pm 6.4$ | $81.7 \pm 1.6$ | $74.9 \pm 4.5$ | 69.4 |
| +T3C | $\mathbf{72.4 \pm 2.0}$ | $\mathbf{54.9 \pm 7.0}$ | $\mathbf{84.3 \pm 1.6}$ | $\mathbf{77.4 \pm 4.4}$ | 72.3 |
| +Tent-BN | $\mathbf{71.4 \pm 2.1}$ | $53.1 \pm 6.7$ | $82.2 \pm 1.5$ | $75.4 \pm 4.4$ | 70.5 |
| +Tent-C | $70.1 \pm 2.3$ | $51.3 \pm 6.4$ | $81.9 \pm 1.6$ | $74.2 \pm 4.7$ | 69.4 |

**(d) TerraIncognita**

|  | L100 | L38 | L43 | L46 | Avg |
|---|---|---|---|---|---|
| ERM | $43.5 \pm 1.6$ | $24.9 \pm 2.0$ | $45.2 \pm 0.2$ | $34.6 \pm 1.0$ | 37.1 |
| +T3C | $\mathbf{44.6 \pm 1.5}$ | $30.1 \pm 1.1$ | $42.6 \pm 0.6$ | $32.8 \pm 1.1$ | 37.5 |
| +Tent-BN | $39.7 \pm 1.4$ | $\mathbf{31.6 \pm 1.1}$ | $40.4 \pm 0.2$ | $33.2 \pm 0.4$ | 36.2 |
| +Tent-C | $\mathbf{45.4 \pm 2.4}$ | $24.0 \pm 2.2$ | $\mathbf{45.4 \pm 0.1}$ | $34.3 \pm 0.9$ | 37.3 |

## B.3 Full Results for Figure 1-a

### B.3.1 Fine-tuning performance

Table 18: Full results for fine-tuning resnet18.

(a) VLCS

| | C | L | S | V | Avg |
|---|---|---|---|---|---|
| ERM | 95.8 ± 0.4 | 60.4 ± 1.1 | 66.4 ± 0.8 | 70.2 ± 1.5 | 73.2 |
| +head | 99.3 ± 0.2 | 70.2 ± 0.9 | 76.6 ± 0.3 | 77.9 ± 0.9 | 81.0 |
| +body | 99.7 ± 0.0 | 72.5 ± 0.8 | 77.8 ± 0.4 | 81.3 ± 0.3 | 82.8 |
| +all | 99.7 ± 0.0 | 72.4 ± 0.8 | 77.8 ± 0.3 | 81.4 ± 0.2 | 82.8 |

(b) PACS

| | A | C | P | S | Avg |
|---|---|---|---|---|---|
| ERM | 78.7 ± 1.3 | 74.3 ± 0.6 | 92.4 ± 0.2 | 75.6 ± 0.8 | 80.3 |
| +head | 86.8 ± 0.3 | 84.7 ± 0.1 | 95.6 ± 0.1 | 86.0 ± 0.3 | 88.3 |
| +body | 90.2 ± 0.3 | 89.7 ± 0.3 | 96.2 ± 0.1 | 91.7 ± 0.1 | 91.9 |
| +all | 90.3 ± 0.4 | 89.7 ± 0.1 | 96.2 ± 0.2 | 91.6 ± 0.0 | 91.9 |

(c) OfficeHome

| | A | C | P | R | Avg |
|---|---|---|---|---|---|
| ERM | 46.1 ± 0.4 | 45.8 ± 0.3 | 65.0 ± 0.3 | 66.0 ± 0.3 | 55.7 |
| +head | 46.1 ± 0.8 | 55.9 ± 0.1 | 74.5 ± 0.1 | 65.5 ± 0.7 | 60.5 |
| +body | 55.1 ± 0.5 | 66.2 ± 0.3 | 78.4 ± 0.3 | 72.0 ± 0.2 | 67.9 |
| +all | 55.0 ± 0.4 | 66.3 ± 0.3 | 78.7 ± 0.4 | 71.8 ± 0.3 | 68.0 |

(d) TerraIncognita

| | L100 | L38 | L43 | L46 | Avg |
|---|---|---|---|---|---|
| ERM | 41.6 ± 3.1 | 36.0 ± 3.0 | 51.0 ± 0.2 | 33.9 ± 1.0 | 40.7 |
| +head | 80.6 ± 0.2 | 75.3 ± 0.3 | 67.9 ± 0.1 | 61.5 ± 1.5 | 71.3 |
| +body | 89.6 ± 0.4 | 86.7 ± 0.5 | 80.6 ± 0.3 | 78.5 ± 0.2 | 83.9 |
| +all | 89.7 ± 0.3 | 86.5 ± 0.1 | 80.9 ± 0.1 | 78.3 ± 0.6 | 83.9 |

Table 19: Full results for fine-tuning resnet50.

(a) VLCS

| | C | L | S | V | Avg |
|---|---|---|---|---|---|
| ERM | 97.6 ± 0.1 | 63.4 ± 0.5 | 69.3 ± 1.6 | 72.0 ± 1.5 | 75.5 |
| +head | 99.1 ± 0.1 | 71.9 ± 0.2 | 78.1 ± 0.3 | 82.5 ± 0.8 | 82.9 |
| +body | 99.9 ± 0.1 | 73.7 ± 0.7 | 80.4 ± 0.7 | 85.1 ± 0.5 | 84.8 |
| +all | 99.7 ± 0.1 | 73.6 ± 0.5 | 80.5 ± 0.6 | 85.1 ± 0.5 | 84.7 |

(b) PACS

| | A | C | P | S | Avg |
|---|---|---|---|---|---|
| ERM | 84.8 ± 0.5 | 78.5 ± 0.6 | 96.1 ± 0.2 | 76.2 ± 1.1 | 83.9 |
| +head | 89.8 ± 0.1 | 88.6 ± 0.8 | 96.4 ± 0.1 | 84.6 ± 0.3 | 89.8 |
| +body | 93.3 ± 0.3 | 93.0 ± 0.5 | 97.5 ± 0.1 | 93.2 ± 0.2 | 94.2 |
| +all | 93.5 ± 0.1 | 93.1 ± 0.5 | 97.4 ± 0.0 | 93.0 ± 0.1 | 94.2 |

(c) OfficeHome

| | A | C | P | R | Avg |
|---|---|---|---|---|---|
| ERM | 58.5 ± 0.4 | 50.7 ± 1.0 | 73.3 ± 0.5 | 75.0 ± 0.2 | 64.4 |
| +head | 58.1 ± 0.5 | 62.1 ± 0.6 | 79.6 ± 0.3 | 73.8 ± 0.7 | 68.4 |
| +body | 67.5 ± 0.8 | 71.7 ± 0.3 | 84.7 ± 0.3 | 79.6 ± 0.1 | 75.9 |
| +all | 67.1 ± 0.8 | 72.5 ± 0.3 | 84.5 ± 0.3 | 79.8 ± 0.3 | 76.0 |

(d) TerraIncognita

| | L100 | L38 | L43 | L46 | Avg |
|---|---|---|---|---|---|
| ERM | 53.4 ± 4.1 | 39.0 ± 1.8 | 52.0 ± 1.0 | 37.1 ± 2.8 | 45.4 |
| +head | 81.4 ± 0.7 | 75.4 ± 1.0 | 67.4 ± 2.1 | 61.4 ± 1.4 | 71.4 |
| +body | 91.0 ± 0.2 | 87.2 ± 0.2 | 80.8 ± 0.6 | 80.0 ± 0.6 | 84.7 |
| +all | 90.2 ± 0.3 | 85.5 ± 1.5 | 81.5 ± 0.3 | 79.9 ± 0.1 | 84.3 |

Table 20: Full results for fine-tuning BiT-M-R50x3.

(a) VLCS

| | C | L | S | V | Avg |
|---|---|---|---|---|---|
| ERM | 93.2 ± 0.4 | 66.1 ± 0.3 | 74.0 ± 0.4 | 73.3 ± 1.2 | 76.7 |
| +head | 98.8 ± 0.3 | 73.3 ± 1.2 | 81.0 ± 0.1 | 83.1 ± 0.9 | 84.1 |
| +body | 99.1 ± 0.2 | 74.3 ± 0.8 | 79.7 ± 0.8 | 83.9 ± 0.6 | 84.2 |
| +all | 99.3 ± 0.2 | 75.4 ± 0.3 | 80.5 ± 0.2 | 84.6 ± 0.4 | 84.9 |

(b) PACS

| | A | C | P | S | Avg |
|---|---|---|---|---|---|
| ERM | 82.0 ± 1.7 | 82.3 ± 0.9 | 95.1 ± 0.4 | 78.1 ± 2.1 | 84.4 |
| +head | 90.1 ± 0.7 | 90.5 ± 0.7 | 97.2 ± 0.2 | 88.5 ± 0.5 | 91.6 |
| +body | 92.0 ± 0.3 | 94.2 ± 0.7 | 96.7 ± 0.1 | 92.9 ± 0.5 | 94.0 |
| +all | 92.5 ± 1.0 | 93.8 ± 0.3 | 96.0 ± 0.5 | 93.0 ± 0.2 | 93.8 |

(c) OfficeHome

| | A | C | P | R | Avg |
|---|---|---|---|---|---|
| ERM | 64.3 ± 1.2 | 56.0 ± 0.9 | 78.1 ± 0.8 | 78.3 ± 0.7 | 69.2 |
| +head | 66.9 ± 0.4 | 68.8 ± 0.7 | 68.3 ± 15.2 | 80.9 ± 0.4 | 71.2 |
| +body | 68.9 ± 2.7 | 72.0 ± 0.6 | 86.6 ± 0.7 | 81.5 ± 0.6 | 77.3 |
| +all | 66.7 ± 1.5 | 73.8 ± 0.7 | 85.4 ± 0.7 | 82.1 ± 0.2 | 77.0 |

(d) TerraIncognita

| | L100 | L38 | L43 | L46 | Avg |
|---|---|---|---|---|---|
| ERM | 60.6 ± 1.4 | 43.5 ± 2.9 | 59.0 ± 0.8 | 46.6 ± 2.9 | 52.5 |
| +head | 86.1 ± 0.5 | 79.9 ± 1.3 | 74.6 ± 0.2 | 72.9 ± 0.7 | 78.4 |
| +body | 91.2 ± 0.4 | 88.6 ± 0.3 | 84.0 ± 0.2 | 81.4 ± 0.2 | 86.3 |
| +all | 91.7 ± 0.0 | 89.0 ± 0.4 | 83.9 ± 0.6 | 82.6 ± 0.8 | 86.8 |

### Table 21: Full results for fine-tuning BiT-M-R101x3.

#### (a) VLCS

|        | C           | L           | S           | V           | Avg  |
|--------|-------------|-------------|-------------|-------------|------|
| ERM    | 93.4 ± 1.4  | 62.7 ± 1.4  | 71.3 ± 0.7  | 72.6 ± 1.3  | 75.0 |
| +head  | **98.5 ± 0.6** | **73.8 ± 1.4** | **79.3 ± 0.9** | **80.2 ± 1.7** | 82.9 |
| +body  | **98.9 ± 0.3** | 72.3 ± 0.3  | **80.6 ± 0.4** | **80.8 ± 0.3** | 83.2 |
| +all   | **99.0 ± 0.1** | **73.6 ± 0.5** | **79.6 ± 1.0** | 79.7 ± 1.2  | 82.9 |

#### (b) PACS

|        | A           | C           | P           | S           | Avg  |
|--------|-------------|-------------|-------------|-------------|------|
| ERM    | 81.1 ± 0.1  | 83.0 ± 0.8  | 94.0 ± 0.5  | 77.7 ± 1.5  | 84.0 |
| +head  | **88.9 ± 0.8** | **90.8 ± 0.2** | **95.8 ± 0.5** | **89.8 ± 0.7** | 91.3 |
| +body  | **91.3 ± 0.6** | **93.6 ± 0.1** | **96.2 ± 0.2** | **93.0 ± 0.5** | 93.5 |
| +all   | 89.2 ± 1.8  | **93.8 ± 0.2** | **95.0 ± 0.6** | **93.1 ± 0.4** | 92.8 |

#### (c) OfficeHome

|        | A           | C           | P           | R           | Avg  |
|--------|-------------|-------------|-------------|-------------|------|
| ERM    | 59.7 ± 1.2  | 55.4 ± 1.1  | 78.7 ± 0.2  | 76.9 ± 0.2  | 67.7 |
| +head  | **63.0 ± 0.7** | **67.0 ± 1.3** | **85.4 ± 0.7** | **79.8 ± 0.4** | 73.8 |
| +body  | **67.9 ± 0.4** | **70.0 ± 1.2** | **82.3 ± 0.6** | 78.7 ± 1.0  | 74.7 |
| +all   | **67.2 ± 0.5** | **72.6 ± 1.1** | **85.2 ± 1.0** | 77.7 ± 2.0  | 75.7 |

#### (d) TerraIncognita

|        | L100        | L38         | L43         | L46         | Avg  |
|--------|-------------|-------------|-------------|-------------|------|
| ERM    | 54.5 ± 4.2  | 39.9 ± 2.6  | 56.5 ± 0.7  | 40.5 ± 0.5  | 47.8 |
| +head  | **84.8 ± 0.8** | **77.4 ± 0.8** | **71.5 ± 0.3** | **69.9 ± 0.8** | 75.9 |
| +body  | **90.9 ± 0.3** | **88.5 ± 0.2** | **82.8 ± 0.5** | **81.9 ± 0.2** | 86.0 |
| +all   | **91.6 ± 0.1** | **88.6 ± 0.2** | **84.5 ± 0.2** | **82.0 ± 0.5** | 86.7 |

### Table 22: Full results for fine-tuning BiT-M-R152x2.

#### (a) VLCS

|        | C           | L           | S           | V           | Avg  |
|--------|-------------|-------------|-------------|-------------|------|
| ERM    | 95.6 ± 1.0  | 66.0 ± 1.0  | 73.3 ± 1.4  | 71.8 ± 0.7  | 76.7 |
| +head  | **99.0 ± 0.2** | **74.2 ± 0.5** | **80.2 ± 0.4** | **80.9 ± 1.0** | 83.6 |
| +body  | **99.6 ± 0.0** | **74.9 ± 0.6** | **81.0 ± 0.3** | **83.8 ± 0.3** | 84.8 |
| +all   | **99.5 ± 0.1** | **74.6 ± 0.3** | **81.3 ± 0.2** | 82.7 ± 1.1  | 84.5 |

#### (b) PACS

|        | A           | C           | P           | S           | Avg  |
|--------|-------------|-------------|-------------|-------------|------|
| ERM    | 80.3 ± 1.4  | 82.2 ± 1.1  | 95.8 ± 0.6  | 82.6 ± 0.2  | 85.2 |
| +head  | **89.4 ± 1.3** | **89.7 ± 0.5** | **96.7 ± 0.2** | **90.7 ± 0.4** | 91.6 |
| +body  | **92.5 ± 0.1** | **95.5 ± 0.2** | **97.2 ± 0.2** | **93.1 ± 0.5** | 94.6 |
| +all   | **92.0 ± 0.0** | **95.2 ± 0.4** | **97.1 ± 0.2** | **93.1 ± 0.6** | 94.4 |

#### (c) OfficeHome

|        | A           | C           | P           | R           | Avg  |
|--------|-------------|-------------|-------------|-------------|------|
| ERM    | 63.2 ± 0.7  | 60.2 ± 1.0  | 79.7 ± 0.6  | 82.2 ± 0.3  | 71.3 |
| +head  | **66.8 ± 2.0** | **71.6 ± 0.7** | **86.6 ± 0.5** | **83.7 ± 0.6** | 77.2 |
| +body  | **72.6 ± 0.7** | **74.1 ± 1.7** | **87.9 ± 0.6** | **83.8 ± 0.2** | 79.6 |
| +all   | **72.3 ± 0.6** | **75.4 ± 1.1** | **86.6 ± 1.2** | 82.8 ± 0.3  | 79.3 |

#### (d) TerraIncognita

|        | L100        | L38         | L43         | L46         | Avg  |
|--------|-------------|-------------|-------------|-------------|------|
| ERM    | 57.3 ± 1.4  | 49.3 ± 1.5  | 58.8 ± 0.5  | 40.1 ± 2.5  | 51.4 |
| +head  | **85.7 ± 0.3** | **78.9 ± 0.7** | **74.2 ± 1.2** | **69.2 ± 1.0** | 77.0 |
| +body  | **91.3 ± 0.1** | **89.1 ± 0.5** | **83.2 ± 0.1** | **82.4 ± 0.2** | 86.5 |
| +all   | **91.2 ± 0.0** | **88.6 ± 0.3** | **82.8 ± 0.3** | **81.8 ± 0.2** | 86.1 |

### Table 23: Full results for fine-tuning ViT-B16.

#### (a) VLCS

|        | C           | L           | S           | V           | Avg  |
|--------|-------------|-------------|-------------|-------------|------|
| ERM    | 97.1 ± 0.5  | 65.4 ± 1.2  | 76.6 ± 0.7  | 77.8 ± 0.5  | 79.2 |
| +head  | **99.2 ± 0.1** | **76.1 ± 0.4** | **84.3 ± 0.3** | **86.7 ± 0.2** | 86.6 |
| +body  | **99.9 ± 0.1** | **78.0 ± 0.5** | **84.1 ± 0.2** | **87.6 ± 0.4** | 87.4 |
| +all   | **99.9 ± 0.1** | **79.0 ± 0.0** | **84.2 ± 0.2** | **87.4 ± 0.3** | 87.6 |

#### (b) PACS

|        | A           | C           | P           | S           | Avg  |
|--------|-------------|-------------|-------------|-------------|------|
| ERM    | 89.9 ± 0.7  | 86.3 ± 0.4  | 99.2 ± 0.1  | 67.2 ± 0.1  | 85.7 |
| +head  | **93.3 ± 0.5** | **92.0 ± 0.4** | **99.2 ± 0.1** | **80.2 ± 0.5** | 91.2 |
| +body  | **95.2 ± 0.2** | **95.3 ± 0.2** | **99.3 ± 0.2** | **88.8 ± 0.5** | 94.7 |
| +all   | **95.2 ± 0.3** | **95.3 ± 0.2** | **99.3 ± 0.2** | **88.7 ± 0.3** | 94.6 |

#### (c) OfficeHome

|        | A           | C           | P           | R           | Avg  |
|--------|-------------|-------------|-------------|-------------|------|
| ERM    | 76.5 ± 0.6  | 62.7 ± 0.2  | 86.9 ± 0.1  | 87.6 ± 0.3  | 78.4 |
| +head  | **78.9 ± 0.3** | **69.3 ± 0.5** | **89.6 ± 0.1** | **88.3 ± 0.6** | 81.5 |
| +body  | **82.0 ± 0.2** | **77.4 ± 0.2** | **92.2 ± 0.2** | **89.9 ± 0.3** | 85.4 |
| +all   | **82.4 ± 0.2** | **77.6 ± 0.2** | **92.1 ± 0.3** | **89.9 ± 0.3** | 85.5 |

#### (d) TerraIncognita

|        | L100        | L38         | L43         | L46         | Avg  |
|--------|-------------|-------------|-------------|-------------|------|
| ERM    | 53.1 ± 1.9  | 26.0 ± 1.0  | 50.3 ± 0.6  | 37.9 ± 0.6  | 41.8 |
| +head  | **84.7 ± 0.1** | **79.2 ± 0.2** | **73.6 ± 0.2** | **70.3 ± 0.4** | 76.9 |
| +body  | **89.6 ± 0.4** | **87.2 ± 0.4** | **81.4 ± 0.0** | **79.7 ± 0.7** | 84.5 |
| +all   | **90.2 ± 0.1** | **87.2 ± 0.3** | **81.5 ± 0.2** | **79.9 ± 0.4** | 84.7 |

### Table 24: Full results for fine-tuning ViT-L16.

#### (a) VLCS

|        | C           | L           | S           | V           | Avg  |
|--------|-------------|-------------|-------------|-------------|------|
| ERM    | 97.7 ± 0.2  | 62.8 ± 1.9  | 74.1 ± 0.2  | 78.1 ± 1.1  | 78.2 |
| +head  | **99.3 ± 0.1** | **77.1 ± 0.4** | **82.4 ± 0.7** | **86.4 ± 0.2** | 86.3 |
| +body  | **99.8 ± 0.1** | **76.7 ± 0.1** | **82.8 ± 0.3** | **87.4 ± 0.3** | 86.7 |
| +all   | **99.7 ± 0.2** | **76.4 ± 0.2** | **82.7 ± 0.3** | **87.1 ± 0.3** | 86.5 |

#### (b) PACS

|        | A           | C           | P           | S           | Avg  |
|--------|-------------|-------------|-------------|-------------|------|
| ERM    | 88.8 ± 0.6  | 83.3 ± 0.5  | 98.6 ± 0.1  | 67.5 ± 2.8  | 84.6 |
| +head  | **93.0 ± 0.3** | **93.2 ± 0.3** | **98.7 ± 0.2** | **83.2 ± 0.4** | 92.0 |
| +body  | **94.8 ± 0.4** | **95.8 ± 0.1** | **98.9 ± 0.2** | **90.5 ± 0.6** | 95.0 |
| +all   | **94.7 ± 0.4** | **96.3 ± 0.2** | **98.9 ± 0.2** | **90.3 ± 1.0** | 95.1 |

#### (c) OfficeHome

|        | A           | C           | P           | R           | Avg  |
|--------|-------------|-------------|-------------|-------------|------|
| ERM    | 77.2 ± 1.2  | 61.6 ± 0.9  | 85.3 ± 1.0  | 87.9 ± 0.4  | 78.0 |
| +head  | **79.1 ± 1.0** | **70.8 ± 0.9** | **89.6 ± 0.8** | **88.9 ± 0.3** | 82.1 |
| +body  | **80.7 ± 0.3** | **80.5 ± 0.4** | **91.8 ± 0.3** | **89.8 ± 0.2** | 85.7 |
| +all   | **80.5 ± 0.1** | **80.2 ± 0.3** | **92.3 ± 0.2** | **89.9 ± 0.1** | 85.7 |

#### (d) TerraIncognita

|        | L100        | L38         | L43         | L46         | Avg  |
|--------|-------------|-------------|-------------|-------------|------|
| ERM    | 56.0 ± 1.3  | 38.9 ± 2.7  | 40.7 ± 8.0  | 35.2 ± 1.6  | 42.7 |
| +head  | **86.2 ± 0.9** | **79.8 ± 0.7** | **75.8 ± 0.4** | **70.9 ± 0.7** | 78.2 |
| +body  | **90.8 ± 0.2** | **87.3 ± 0.9** | **81.5 ± 0.7** | **79.5 ± 0.4** | 84.8 |
| +all   | **90.6 ± 0.4** | **86.9 ± 0.2** | **81.7 ± 0.2** | **78.9 ± 0.7** | 84.5 |

Table 25: Full results for fine-tuning DeiT.

(a) VLCS

|  | C | L | S | V | Avg |
|---|---|---|---|---|---|
| ERM | 97.3 ± 0.4 | 64.0 ± 0.6 | 76.9 ± 0.6 | 78.9 ± 0.8 | 79.3 |
| +head | **99.7 ± 0.2** | **72.1 ± 0.2** | **82.2 ± 0.7** | **86.8 ± 0.6** | 85.2 |
| +body | **99.9 ± 0.0** | **77.3 ± 0.3** | **85.0 ± 0.2** | **88.4 ± 0.4** | 87.7 |
| +all | **99.9 ± 0.0** | **77.3 ± 0.3** | **85.0 ± 0.2** | **88.4 ± 0.4** | 87.7 |

(b) PACS

|  | A | C | P | S | Avg |
|---|---|---|---|---|---|
| ERM | 92.8 ± 0.5 | 83.8 ± 0.5 | 98.5 ± 0.1 | 76.2 ± 1.4 | 87.8 |
| +head | **95.9 ± 0.1** | **94.1 ± 0.4** | **99.2 ± 0.1** | **91.5 ± 0.5** | 95.2 |
| +body | **96.6 ± 0.4** | **96.0 ± 0.1** | **99.3 ± 0.1** | **94.6 ± 0.2** | 96.6 |
| +all | **96.6 ± 0.4** | **96.1 ± 0.1** | **99.3 ± 0.1** | **94.6 ± 0.2** | 96.6 |

(c) OfficeHome

|  | A | C | P | R | Avg |
|---|---|---|---|---|---|
| ERM | 74.5 ± 0.2 | 63.3 ± 0.6 | 83.2 ± 0.2 | 85.5 ± 0.2 | 76.6 |
| +head | **78.6 ± 0.3** | **75.4 ± 0.1** | **89.5 ± 0.0** | **87.3 ± 0.2** | 82.7 |
| +body | **80.9 ± 0.0** | **79.7 ± 0.2** | **91.0 ± 0.2** | **88.4 ± 0.1** | 85.0 |
| +all | **80.9 ± 0.0** | **79.7 ± 0.2** | **91.0 ± 0.2** | **88.5 ± 0.1** | 85.0 |

(d) TerraIncognita

|  | L100 | L38 | L43 | L46 | Avg |
|---|---|---|---|---|---|
| ERM | 57.5 ± 2.9 | 44.0 ± 2.0 | 56.0 ± 0.4 | 42.3 ± 0.8 | 50.0 |
| +head | **88.5 ± 0.3** | **84.5 ± 0.1** | **79.3 ± 0.5** | **75.6 ± 0.2** | 82.0 |
| +body | **90.8 ± 0.4** | **88.9 ± 0.3** | **83.6 ± 0.3** | **82.1 ± 0.1** | 86.3 |
| +all | **90.8 ± 0.4** | **88.7 ± 0.0** | **83.6 ± 0.1** | **81.7 ± 0.4** | 86.2 |

Table 26: Full results for fine-tuning HViT.

(a) VLCS

|  | C | L | S | V | Avg |
|---|---|---|---|---|---|
| ERM | 96.8 ± 0.5 | 64.1 ± 0.9 | 75.9 ± 1.1 | 80.0 ± 1.2 | 79.2 |
| +head | **99.6 ± 0.1** | **76.9 ± 1.0** | **83.1 ± 0.2** | **87.3 ± 0.8** | 86.7 |
| +body | **99.8 ± 0.0** | **77.0 ± 0.2** | **84.1 ± 0.3** | **88.7 ± 0.3** | 87.4 |
| +all | **99.8 ± 0.0** | **77.2 ± 0.3** | **84.7 ± 0.4** | **88.5 ± 0.4** | 87.6 |

(b) PACS

|  | A | C | P | S | Avg |
|---|---|---|---|---|---|
| ERM | 89.5 ± 1.4 | 85.9 ± 2.6 | 98.0 ± 0.5 | 85.5 ± 1.1 | 89.7 |
| +head | **93.1 ± 0.5** | **93.7 ± 0.6** | **98.6 ± 0.3** | **92.3 ± 0.0** | 94.4 |
| +body | **96.0 ± 0.2** | **97.3 ± 0.4** | **99.0 ± 0.1** | **95.4 ± 0.2** | 96.9 |
| +all | **96.0 ± 0.3** | **97.2 ± 0.4** | **98.9 ± 0.1** | **95.3 ± 0.0** | 96.8 |

(c) OfficeHome

|  | A | C | P | R | Avg |
|---|---|---|---|---|---|
| ERM | 77.3 ± 1.2 | 68.4 ± 0.6 | 87.0 ± 0.2 | 87.5 ± 0.3 | 80.0 |
| +head | **80.4 ± 1.4** | **77.0 ± 0.4** | **90.3 ± 0.1** | **89.1 ± 0.1** | 84.2 |
| +body | **82.6 ± 0.9** | **83.3 ± 0.4** | **92.7 ± 0.3** | **89.9 ± 0.3** | 87.1 |
| +all | **82.7 ± 0.9** | **83.2 ± 0.5** | **93.0 ± 0.1** | **89.7 ± 0.4** | 87.2 |

(d) TerraIncognita

|  | L100 | L38 | L43 | L46 | Avg |
|---|---|---|---|---|---|
| ERM | 62.3 ± 2.2 | 44.6 ± 0.5 | 56.6 ± 0.6 | 41.9 ± 1.0 | 51.4 |
| +head | **88.8 ± 0.5** | **83.9 ± 0.1** | **79.0 ± 0.7** | **75.3 ± 0.3** | 81.8 |
| +body | **91.4 ± 0.4** | **89.7 ± 0.2** | **83.0 ± 0.8** | **81.8 ± 0.6** | 86.5 |
| +all | **91.8 ± 0.1** | **89.2 ± 0.1** | **82.4 ± 0.5** | **82.4 ± 0.2** | 86.4 |

Table 27: Full results for fine-tuning Mixer-L16.

(a) VLCS

|  | C | L | S | V | Avg |
|---|---|---|---|---|---|
| ERM | 98.8 ± 0.2 | 61.0 ± 0.4 | 72.5 ± 0.9 | 73.5 ± 0.3 | 76.4 |
| +head | **99.6 ± 0.1** | **72.8 ± 0.4** | **80.6 ± 0.5** | **83.7 ± 0.4** | 84.2 |
| +body | **99.7 ± 0.1** | **73.9 ± 0.2** | **81.2 ± 0.4** | **84.2 ± 0.3** | 84.8 |
| +all | **99.6 ± 0.2** | **74.2 ± 0.1** | **81.0 ± 0.4** | **83.7 ± 0.6** | 84.6 |

(b) PACS

|  | A | C | P | S | Avg |
|---|---|---|---|---|---|
| ERM | 79.9 ± 3.2 | 80.3 ± 0.8 | 97.6 ± 0.4 | 67.5 ± 0.3 | 81.3 |
| +head | **89.6 ± 1.2** | **90.9 ± 0.5** | **98.5 ± 0.1** | **81.5 ± 0.8** | 90.1 |
| +body | **90.4 ± 0.7** | **93.6 ± 0.3** | 96.6 ± 0.5 | **86.3 ± 0.5** | 91.7 |
| +all | **90.8 ± 0.7** | **93.7 ± 0.4** | 96.5 ± 0.5 | **86.7 ± 1.7** | 91.9 |

(c) OfficeHome

|  | A | C | P | R | Avg |
|---|---|---|---|---|---|
| ERM | 69.9 ± 2.4 | 51.3 ± 6.4 | 81.7 ± 1.6 | 74.9 ± 4.5 | 69.4 |
| +head | **74.3 ± 1.7** | **67.0 ± 4.7** | **88.4 ± 0.8** | 79.0 ± 3.7 | 77.2 |
| +body | **73.5 ± 0.6** | **71.3 ± 4.2** | **87.7 ± 0.5** | **81.1 ± 2.9** | 78.4 |
| +all | **73.2 ± 0.7** | **71.7 ± 4.2** | **88.2 ± 0.4** | **80.3 ± 2.6** | 78.4 |

(d) TerraIncognita

|  | L100 | L38 | L43 | L46 | Avg |
|---|---|---|---|---|---|
| ERM | 43.5 ± 1.6 | 24.9 ± 2.0 | 45.2 ± 0.2 | 34.6 ± 1.0 | 37.1 |
| +head | **87.5 ± 0.3** | **79.9 ± 0.9** | **76.3 ± 0.6** | **67.6 ± 0.4** | 77.8 |
| +body | **90.1 ± 0.2** | **85.4 ± 0.2** | **80.3 ± 0.1** | **76.7 ± 0.3** | 83.1 |
| +all | **89.8 ± 0.1** | **85.6 ± 0.2** | **80.8 ± 0.3** | **76.6 ± 0.9** | 83.2 |

## B.3.2 Prediction entropy on seen and unseen domains

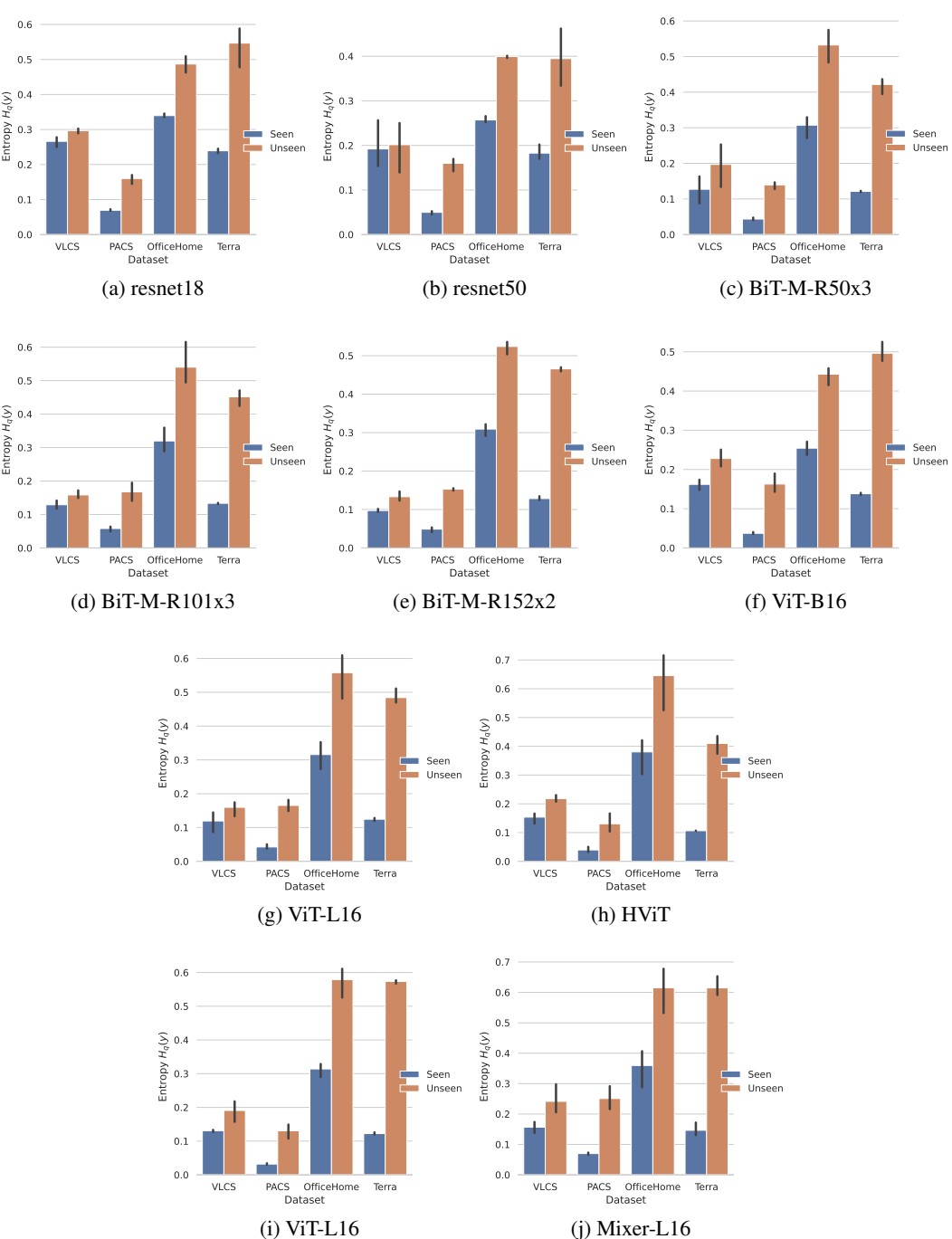

Figure 3: Full results for Figure 1-b. Each figure corresponds to the results of different backbone networks.

## B.4 Tent vs. T3A in entropy

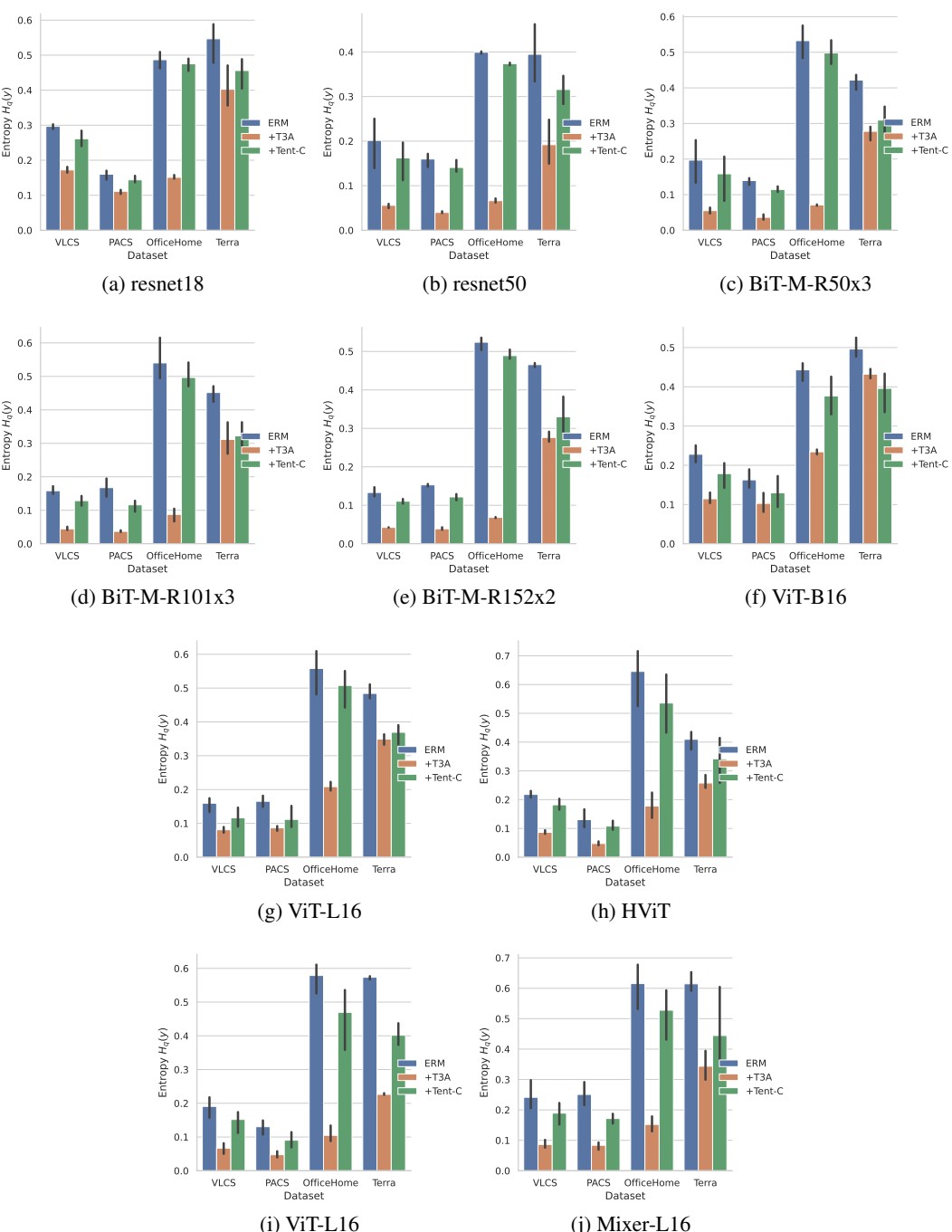

Figure 4: Full results for Figure 1-c. Each figure corresponds to the results of different backbone networks.

# C Comparing T3A with Other Test-Time Adaptation Methods using Oracle Model Selection

Table 28: Comparison of our method and existing test-time adaptation methods. Unlike Table 3, the results here select the hyper-parameter in the test-domain validation set (Oracle setup [17]). As with Table 3, this experiments is conducted only on the default hyperparameters of ERM. Bold type indicates performance improvement, and * indicates statistical significance in paired t-test (* indicates $p \leq 0.05$).

| Models | VLCS | PACS | OfficeHome | Terra | Avg |
|---|---|---|---|---|---|
| resnet18-BN | $73.0 \pm 0.6$ | $79.5 \pm 0.4$ | $61.8 \pm 0.3$ | $41.7 \pm 0.9$ | 64.0 |
| SHOTIM | $60.8 \pm 0.3$ | $\mathbf{81.5 \pm 0.2}$ | $\mathbf{62.4 \pm 0.3}$ | $29.5 \pm 1.3$ | 58.6 |
| SHOT | $61.2 \pm 0.4$ | $\mathbf{81.4 \pm 0.2}$ | $\mathbf{62.4 \pm 0.3}$ | $32.3 \pm 0.6$ | 59.3 |
| PseudoLabel | $65.4 \pm 0.2$ | $73.1 \pm 2.5$ | $54.7 \pm 4.6$ | $37.7 \pm 2.5$ | 57.7 |
| PLClf | $72.5 \pm 1.3$ | $78.4 \pm 0.8$ | $61.8 \pm 0.3$ | $\mathbf{43.9 \pm 1.3}$ | 64.2 |
| TentFull | $\mathbf{73.8 \pm 0.8}$ | $\mathbf{84.7 \pm 0.2}$ | $\mathbf{62.7 \pm 0.1}$ | $37.4 \pm 0.8$ | 64.7 |
| TentNorm | $70.3 \pm 0.9$ | $\mathbf{82.7 \pm 0.1}$ | $\mathbf{62.1 \pm 0.1}$ | $36.6 \pm 0.2$ | 62.9 |
| TentClf | $72.3 \pm 1.1$ | $77.3 \pm 1.8$ | $61.3 \pm 0.3$ | $37.8 \pm 2.7$ | 62.2 |
| TentPreBN | $64.7 \pm 0.7$ | $\mathbf{81.2 \pm 0.2}$ | $\mathbf{62.6 \pm 0.3}$ | $36.4 \pm 1.0$ | 61.2 |
| T3A | $\mathbf{73.8 \pm 0.8}$ | $\mathbf{81.3 \pm 0.0}$ | $\mathbf{62.8 \pm 0.3}$ | $40.5 \pm 0.3$ | 64.6* |
| resnet50-BN | $74.3 \pm 0.5$ | $84.1 \pm 0.1$ | $66.9 \pm 0.2$ | $45.8 \pm 1.8$ | 67.8 |
| SHOTIM | $55.2 \pm 4.6$ | $83.7 \pm 0.2$ | $\mathbf{67.5 \pm 0.2}$ | $27.0 \pm 1.4$ | 58.3 |
| SHOT | $61.4 \pm 1.5$ | $\mathbf{85.1 \pm 0.4}$ | $\mathbf{67.5 \pm 0.3}$ | $30.9 \pm 2.3$ | 61.2 |
| PseudoLabel | $64.2 \pm 2.3$ | $70.4 \pm 1.0$ | $55.0 \pm 1.8$ | $33.0 \pm 7.4$ | 55.6 |
| PLClf | $73.5 \pm 0.9$ | $\mathbf{84.6 \pm 0.4}$ | $66.7 \pm 0.2$ | $\mathbf{46.7 \pm 1.6}$ | 67.9 |
| TentFull | $\mathbf{74.8 \pm 1.2}$ | $\mathbf{87.7 \pm 0.1}$ | $\mathbf{67.0 \pm 0.4}$ | $42.9 \pm 0.2$ | 68.1 |
| TentNorm | $71.4 \pm 0.5$ | $\mathbf{85.7 \pm 0.1}$ | $66.6 \pm 0.0$ | $42.4 \pm 0.4$ | 66.5 |
| TentClf | $72.6 \pm 0.8$ | $83.3 \pm 0.5$ | $66.7 \pm 0.2$ | $45.0 \pm 0.9$ | 66.9 |
| TentPreBN | $65.7 \pm 1.4$ | $\mathbf{84.9 \pm 0.0}$ | $\mathbf{67.7 \pm 0.2}$ | $42.7 \pm 0.5$ | 65.3 |
| T3A | $\mathbf{76.1 \pm 0.2}$ | $\mathbf{85.2 \pm 0.2}$ | $\mathbf{67.8 \pm 0.2}$ | $\mathbf{46.0 \pm 1.5}$ | 68.8** |

# D    ImageNet Top-1 Accuracy vs. DG Performance

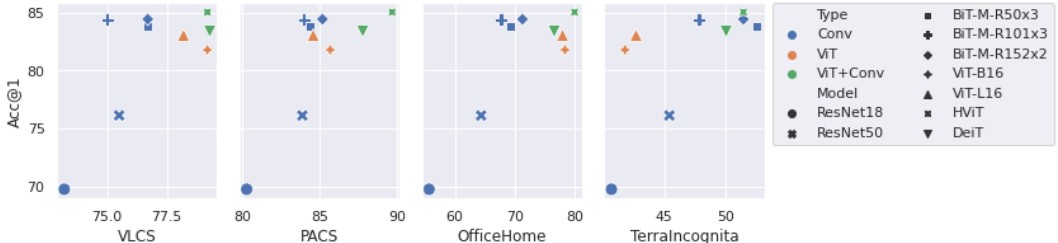

(a) ImageNet top-1 accuracy vs. domain generalization performance.

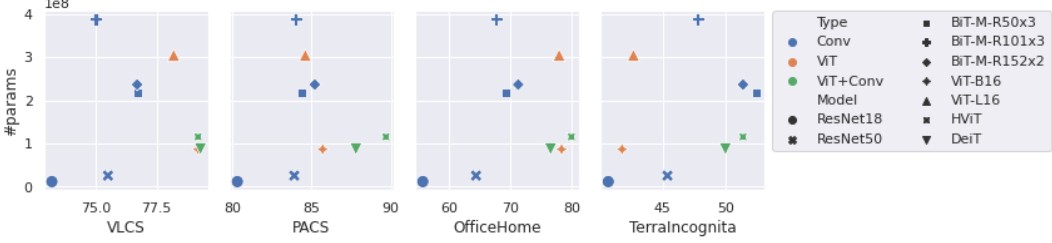

(b) #Parameters vs. domain generalization performance.

Figure 5: Comparing domain generalization performance on four datasets and properties of the pretrained models. (a) ImageNet top-1 accuracy of each backbone networks vs. domain generalization performance. (b) The number of parameters of each backbone networks vs. domain generalization performance.

Figure 5 illustrates the correlation between domain generalization performance on each dataset and properties of the backbone networks (ImageNet top-1 accuracy and the number of parameters). For visualization, we split backbone networks into (1) Conv (blue): pure convolution (ResNet18, ResNet50, and BiT), (2) ViT (orange): pure vision transformer (ViT-B16 and ViT-L16), and (3) ViT+Conv (green): hybrid of convolution and transformer (HViT and DeiT). Note that, DeiT does not use convolution in the architecture design, but it distills the knowledge from the convolution-based model during training. Each marker corresponds to different backbone networks.

We can make the following observations. (1) Looking only at Conv, there is a correlation between the performance of the ImageNet and the performance of DG, but there is no correlation across architectures. (2) ViT+Conv models work well on all datasets. ViT models also significantly outperform Conv models in VLCS and OfficeHome but perform significantly worse in TerraIncognita datasets. (3) There is no correlation between the number of parameters and domain generalization performance. While most literature in DG focuses on ResNet50, this result suggests the practical importance of choosing the network structure according to the datasets and methods that can improve the performance regardless of the model.