# OpenReview forum: "Test-Time Classifier Adjustment Module for Model-Agnostic Domain Generalization"
_NeurIPS.cc/2021/Conference — NeurIPS 2021 Spotlight_

### Official Review · Reviewer_rrrY · 2021-07-15

**Rating:** 7
**Confidence:** 5

**Summary:**

This paper proposed a test-time adaptation (TTA) solution for domain generalization (DG) problems. It exploits prototypical representations, which are used to classify test samples and are refined over time. The method performs well, when compared with standard DG methods and the recently introduced TTA method "TENT".

**Limitations And Societal Impact:**

Yes.

**Main Review:**

Strengths

- The method proposed method is reasonable, and scores favorably with competing algorithms in standard DG benchmarks.

- The paper is overall well written, with Figures well designed and properly placed.

- The hyper-parameter selection is transparent, also for what concerns the competing algorithm. The plots in Figure 2 are very helpful in assessing the robustness of the proposed method with respect to the baseline across different runs.

Weaknesses

- According to Lines 167/168, the prototypes are refined every time a new test sample is processed: this is consistent with Wang et al. ICLR 2021 (TENT), but it brings an advantage over standard DG methods that the method is compared against - which make their decision by evaluating single samples independently. It would be good to include also some results where the prototypes are not updated, to have a fairer comparison with DG methods. For what concerns the protocol followed in this work and by Wang et al. ICLR 2021, it would be good to also include a comparison against Schneider et al. ICLR 2021 ("Improving robustness against common corruptions by covariate shift adaptation"): it is essentially a simpler version of TENT, where only the BN statistics is updated.

- I believe the comparison with TENT (and with the suggested Schneider et al., ICLR 2021) should be carried out by using a standard ResNet with BN: I am not fully convinced that the precautions taken (lines 246/247) put the competing algorithms in a favorable starting point. It should be fairly simple to test the proposed method with a ResNet-18/50 with BN, am I correct? In this way, the comparison with such related work (the most related to the proposed method) would utmost fair.

Additional comments

- Line 162: why is the support set initialized with the weights of the network, and not with the average between the source features? In Eq.3 new features are included in the support set, and in Eq.5 the average between the elements of the support set is carried out: from the latter Eq., it seems that they are all feature vectors, without weight vectors. Is it a typo or am I getting this wrong?

- The motivation in lines 126-129 are not very strong: it is true that test-time adaptation is yet to be used in DG, but it was tested in very related problems - for instance, under corrupted test samples or natural domain shifts.

- In the related work, it should be included [Chidlovskii et al., "Adaptation in the Absence of Source Domain Data", SIGKDD 2016] for what concerns source-free adaptation, which I believe it is the first work tackling this problem formulation.

- Researchers started exploiting prototypical representations for classification before they were used in the cited few-shot learning works. See, e.g., Mensink et al., "Distance-based image classifi- cation: Generalizing to new classes at near-zero cost" TPAMI 2013.

- Lines 253-258, I think [47] should be [16]

- I do not believe that Remark 1  in Sec.3.2 is relevant: it is widely accepted that finetuning the final layer carries good performance in general.

**Time Spent Reviewing:**

/

---

> ### Author Response · Authors · 2021-08-10
> **Reply to Reviewer rrrY**
>
> We thank the reviewer for the valuable feedback. Please find our responses below.
>
> ---
>
> **Comparison with the actual TENT and its variant.**
> First of all, we sincerely thank the reviewer for thoroughly reading our paper and provide detailed comments. Following the reviewer's comments, we have conducted additional experiments, including new baselines suggested by the reviewer.  Since the amount of content is large and the comments are common to other reviewers, we will mention the results of this experiment in another thread, so please check that thread for details. In short, the results confirm that:
>
> 1. T3A stills work better than the actual Tent and updating BN statics [a], which the reviewer suggests.
> 2. The easiness of the model selection (robustness to the hyperparameter) is one of the merits of the proposed method over the existing optimization-based method.
> 3. The common practice in SFDA (offline setup) does not work well in an online setup.
>
> It is worth mentioning that the reason why we did not test the actual Tent in the first manuscript is that it is the common practice of DG to fix (remove) the BN layers of ResNet before the fine-tuning on source domains (see appendix D of [16]). The performance reported in [16] referred to in our paper does not use any BN layer during either training and testing. Thus, it is difficult to directly compare the performance with these methods if we use the ResNet w/ BN as backbone networks. We plan to include the additional results while keeping the current contents and discuss the (1) comparison to the DG method and (2) comparison to existing (online) adaptation methods independently.
>
> [a] Schneider, Steffen et al. “Improving robustness against common corruptions by covariate shift adaptation.” NeurIPS2020
>
> **Offline adaptation setup.**
> Speaking of offline setup, we agree that the adaptation brings advantages to our method compared to DG. However, this is indeed the central claim of our paper, and therefore can not be changed. Besides, if we test our method in an offline setting, it would make the comparison with the existing DG method more unfair because it would use additional unlabeled data from the target distribution that is not assumed in the standard DG method. If you have further questions or concerns on this point, we are very welcome to hear that.
>
> **Why is the support set initialized with the weights of the network?**
> We are sorry for confusing you. As the reviewer mentioned, we can initialize the support set with the source features. However, we initialize it with the weight of the last linear layer since it can be naturally considered the templates of each class on the source domains. For example, assume the N is the feature dimension, and K is the number of classes. Then the last linear layer contains N by K weight matrix, and each N-dimensional vector can be regarded as the template for each category (up to the scale differences). We, therefore, normalize it and directly use it as the initialization of the support set. In addition to the computational simplicity, our procedure can work with source-free settings, where one can only access the model parameters but can not access source data itself due to privacy reasons. We will clarify this point in the final version.
>
> **The motivation in lines 126-129 are not very strong.**
> The difference between the prior test-time adaptation is that our method is optimization-free; it does not use SGD during testing. The additional experimental results suggest that the optimization-free method has an advantage over optimization-based methods since it is more robust to hyperparameters. In addition, the proposed method has advantages over many existing studies in terms of inference speed, as discussed in the current manuscript. Although our paper primarily aims to provide an insight into the DG community by testing a new approach, it is worth noting that our method can work well in other settings due to these advantages. We will emphasize this point in the final manuscript.
>
> **Missing literature.**
> Thank you for providing the missing literature. We will add it to the final version.
>
> Regarding [b], the primal difference between [b] and our work is that [b] assumes the existence of several labeled data from the target data similar to the prototypical networks [39]. More generally speaking, several methods in incremental learning (e.g., [b, c]) use the supervised prototypes. As discussed in section 2.2, we combine prototypical networks with pseudo-labeling techniques to handle the absence of labeled data from the target distribution. We will add the discussion to the final manuscript.
>
> [b] Mensink, Thomas et al. “Distance-Based Image Classification: Generalizing to New Classes at Near-Zero Cost.” TPAMI (2013)
>
> [c] Rebuffi, Sylvestre-Alvise et al. “iCaRL: Incremental Classifier and Representation Learning.” CVPR (2017)
>
> **Lines 253-258, I think [47] should be [16].**
> Thank you for pointing out the typos. We will fix it following the comment.
>
>
> **Remark 1 in Sec.3.2.**
> We agree that the experimental finding in Sec.3.2 itself is not revolutionary. We added this experiment because most existing works focus on obtaining a good feature extractor rather than a classifier as with our method. For example, the majority of papers focus on learning domain-invariant representation. We added this section to emphasize the importance of the classifier itself even in DG setup, rather than to modulate the feature extractor as with most prior works.
>
> In summary, we still believe that the content is worth remaining, especially for the DG community.  However, we will shorten the text given that fine-tuning in the final layer generally works.
>
> ---
> Again, thank you for giving us the opportunity to strengthen our manuscript with your valuable comments and queries. We will modify the manuscript following the above discussion.

---

> > ### Comment · Reviewer_rrrY · 2021-08-12
> > **Thank you for your response**
> >
> > Thank you for replying to my feedback.
> >
> > The experiments carried out by the Authors are convincing, and add significant value to the paper. I hope they will be included in the final manuscript, at least in the Appendix: they make the comparison against approaches that rely on batch-norm significantly fairer.
> >
> > For what concerns the leverage over standard DG methods, I understand that this cannot be changed. Yet, it should be stressed somewhere in the manuscript that the proposed approach, overall, makes use of more data to make its final predictions - when compared with DG approaches. Similar discussions are reported in Schneider et al. NeurIPS 2020, where - in other words - the Authors claim that not considering the whole test distribution is an unnecessarily difficult problem, since we typically have multiple samples.
> >
> > I am happy to increase my score - I am positive that the revised version of this work can be interesting for the NeurIPS community.

---

> > > ### Author Response · Authors · 2021-08-24
> > > **Thank you for reevaluating our work**
> > >
> > > We thank the reviewer for taking our response into account. We will definitely include the experiments and discussion in the final manuscripts. In addition, I will clarify the potential benefits of our method compared with DG approaches following the reviewer's comment.

---

### Official Review · Reviewer_mpea · 2021-07-16

**Rating:** 7
**Confidence:** 5

**Summary:**

This paper proposes a new online method for test-time adaptation to domain shift. Specifically, the final layer of a network (i.e. the classifier) is adapted to test data using pseudo-prototypes, which combine the ideas of pseudo-labelling [1] and prototypical representations [2]. That is, pseudo-labels define the support set for each class, and the centroids of these support sets are the prototypes. Experiments demonstrate incremental but reliable performance improvements in the test domain using this optimization-free method.

**Limitations And Societal Impact:**

Yes, the authors have addressed the limitations and potential ethical concerns.

**Main Review:**

**Summary**: Online test-time domain *adaptation* is an interesting and useful research direction, overcoming many of the impracticalities of domain *generalization* (no updating on the test data). The authors’ combination of two existing ideas -- pseudo-labelling [1] and prototypical representations [2] -- seem effective for this setting. The improvements are incremental but consistent. I believe the paper is just above the acceptance threshold, and would be improved with the following: (i) stronger baselines (see below); and (ii) a comparison to “offline” optimization-based methods [3,4] when using two different model selection strategies---training-domain validation-set and test-domain validation-set (oracle) validation sets (see below).

**Originality**: A new combination of well-known techniques for the purpose of *online* test-time adaptation. Related work is adequately cited, making clear how this work differs from source-free domain adaptation (offline vs. online) and prototypical networks (assume access to a few labels in the test domain vs. pseudo-labels).


**Quality**: Overall, the submission is technically sound, the claims well supported, and the methods appropriate. However, the empirical evaluation could be improved with:
- *Additional baselines*
     - *Full Tent method*: Tent-C and Tent-BN are very weak baselines -- they are quite far from the proposed Tent method [4]. In addition, many networks *do* still use batch normalization, including some of the backbones used in this paper (e.g. ResNet50). Thus I think a comparison to the actual Tent method (i.e. updating all BN parameters) is warranted, e.g. for the ResNet models. I think the difficulty of model selection will ensure the proposed method still “wins”, but the comparison is important.
     - *SFDA methods in an online setting*: Do they do poorly? They seem like necessary baselines to show the advantages of the proposed method (e.g. better sample-efficiency due to less updates, or better performance due to easier model selection). E.g. use SHOT[3] with 200 examples per test batch, rather than the current 32. This is the online setting of Tent [4].
     - *Pseudo-labelling*: could the final layer be updated with pseudo-labelling to isolate the benefits of: (i) prototypes vs. pseudo-labelling; and (ii) being optimization-free (easier model selection)?
- *Model selection*: This is a huge advantage of the proposed method over optimization-based methods (e.g. Tent or SFDA methods). Thus, I think it warrants a little more emphasis / analysis:
     - *Lines 261--262*: \emph{These parameters must be selected before the deployment}
     - Compare the performance of your method to these optimization-based methods when using: (i) training-domain validation-set; and (ii) test-domain validation set (oracle). This would very clearly illustrate the limitations of these methods, and thus the advantage of the proposed one. E.g. a figure similar to Figure 1a for different model selection methods.
     - *Figure 1c*: I think this figure is a bit unfair/misleading as: (i) Tent-C is used, rather than the actual Tent method[4], despite ResNet50 having BN layers; and (ii) the poor entropy-reduction of Tent-C is likely explained by poor model selection (training-domain validation set), and this point is only very briefly noted in lines 208--209---not in the caption.

**Clarity**: The submission is clearly written and well organized.

**Significance**: The importance and usefulness is relatively clear. The proposed method improves the state-of-the-art for *online* test-time adaptation on networks with *no batch-normalization*. The significance and usefulness would be much improved by: (i) comparing to the actual Tent method [4] on the resnet models which use BN; and (ii) illustrating the poor suitability of existing *offline* SFDA methods to this “online” problem, and as a result, the need for the proposed method.

**Additional feedback**:
- *A model can be confident but wrong.* The authors seem to assume that this never happens. E.g. on line 128, “more confidence predictions are more correct” -- this needs to be weakened, e.g. “more confident predictions are often more correct”... but not always. Adversarial examples are one… example, but I’m sure it can happen even when not hand-crafted. I think this is a minor limitation worth acknowledging (also applies to other entropy-reduction methods too).
- *Domain generalization vs. domain adaptation.* I am not sure I agree with calling this a domain *generalization* method, rather than an online method for source-free domain *adaptation*. DG usually implies *no updating* on the test data.



[1] Lee, D. H. (2013). Pseudo-label: The simple and efficient semi-supervised learning method for deep neural networks. In *Workshop on Challenges in Representation Learning, ICML*, 3, pp. 896.

[2] Snell, J., Swersky, K., & Zemel, R. (2017). Prototypical Networks for Few-shot Learning. In *Advances in Neural Information Processing Systems*, 30, pp. 4077-4087.

[3] Liang, J., Hu, D., & Feng, J. (2020). Do we really need to access the source data? Source hypothesis transfer for unsupervised domain adaptation. In *International Conference on Machine Learning*, pp. 6028-6039.

[4] Wang, D., Shelhamer, E., Liu, S., Olshausen, B., & Darrell, T. (2020). Tent: Fully Test-Time Adaptation by Entropy Minimization. In *International Conference on Learning Representations*.


**Time Spent Reviewing:**

4

---

> ### Author Response · Authors · 2021-08-10
> **Reply to Reviewer mpea**
>
> We thank the reviewer for the valuable feedback. Please find our responses below.
>
> ---
>
> **Additional baselines and model selection.**
> First of all, we sincerely thank the reviewer for thoroughly reading our paper and provide detailed comments. Following the reviewer comments, we have conducted additional experiments, including new baselines and different model selection strategies as suggested by the reviewer. Since the amount of content is large and the comments are common to other reviewers, we will mention the results of this experiment in another thread, so please check that thread for details. In short, the results confirm that:
>
> 1. T3A stills work better than the actual Tent.
> 2. The easiness of the model selection (robustness to the hyperparameter) is one of the merits of the proposed method over the existing optimization-based method.
> 3. The common practice in SFDA (offline setup) does not work well in an online setup.
>
> It is worth mentioning that the reason why we did not test the actual Tent in the first manuscript is that it is the common practice of DG to fix (remove) the BN layers of ResNet before the fine-tuning on source domains (see appendix D of [16]). The performance reported in [16], which is referred to in our paper, does not use any BN layer during either training and testing. Therefore, it is difficult to directly compare the performance with these methods if we use the ResNet w/ BN as backbone networks. We, therefore, plan to include the additional results while keeping the current results and discuss the (1) comparison to DG method and (2) comparison to existing (online) adaptation methods independently.
>
> **Figure 1-c.**
> Thank you for raising the concerns. We agree that the current description is a bit misleading and fix it in the final manuscript by either (1) emphasizing the possibility of poor model selection (training-domain validation set) for Tent-C in the caption or (2) replacing it with the results of Tent-Full.
>
> **A model can be confident but wrong.**
> We agree with the comments and are sorry for making a too bold statement in the current manuscripts. We acknowledge that many papers discuss the model calibration and how softmax entropy fails to characterize the performance in practice. We will mild the sentence (maybe "more confidence predictions tend to be correct") and discuss the possibility of other more appropriate metrics to characterize the model performance.
>
> **Domain generalization vs. domain adaptation.**
> Thank you for giving an interesting suggestion. We use the term "domain generalization" throughout the paper because this work is primarily motivated by the recent work [16], which shows various DG algorithms fail to give significant performance gain compared to the simple ERM. We try to provide insight into this community by testing an unexplored approach in the DG setup (focusing on the test phase rather than the training phase). However, we agree that we can frame our method as an "online" and source-free domain transfer method and verify it on domain generalization setup. We will add the discussion in the final manuscript. In addition, if all reviewers think it is better to emphasize this point, we can slightly change the title to highlight it (e.g., "Test-Time Template Adjustments Module for Online Source-free Domain Transfer").
>
> ---
> Again, thank you for giving us the opportunity to strengthen our manuscript with your valuable comments and queries. We will modify the manuscript following the above discussion.

---

> > ### Comment · Reviewer_mpea · 2021-08-20
> > **Response to authors**
> >
> > Dear authors,
> >
> > Thank you for your detailed response and additional experiments. I have raised my score by 1 in response, as I believe many of my concerns were addressed.
> >
> > Two points/concerns/questions:
> >  1. I am quite surprised by the poor reported performance of SHOT. Did you use the full method from the paper, including label smoothing when training on the source domain? My main surprises/concerns were:
> >     - ~4% lower on Office-Home than reported in the original paper.
> >     - Performance gets worse (than the unadapted model) even when using samples from the test domain to set hyperparameters—surely the performance must be at least as good since one could just set LR=0 and match the performance?
> >  2. *"The common practice in SFDA (offline setup) does not work well in an online setup"* —what is the common practice in SFDA (updating most parameters?)? And offline vs. online here refers to multiple vs. single pass through the test data, correct?

---

> > > ### Author Response · Authors · 2021-08-24
> > > **Additional response**
> > >
> > > We thank the reviewer for taking our response into account.  Below are responses to the additional questions.
> > >
> > > ---
> > >
> > > > Did you use the full method from the paper, including label smoothing when training on the source domain?
> > >
> > > Thank you for the detailed questions. We did not use the label-smoothing when training on the source domain. Here, we focus on the adaptation method, and therefore the source model is the same as the other baselines for the fair comparison. We will clarify it in the final version for the potential causes of performance degradation in SHOT and SHOT-IM.
> > >
> > > > ~4% lower on Office-Home than reported in the original paper.
> > >
> > > The experimental setting in the original paper [1] is different from our setup. Therefore we can not directly compare it. More specifically, the results in [1] are the domain adaptation setup, where one domain is selected as the source domain, and another domain is selected as the target domain (4*3=12 setups in total). On the other hand, our setup is a domain generalization setup, where one domain is selected as the target domain, and the remaining domains are used as the source domains (4 setups in total).
> > >
> > > > Performance gets worse (than the unadapted model) even when using samples from the test domain to set hyperparameters—surely the performance must be at least as good since one could just set LR=0 and match the performance?
> > >
> > > I think there have several reasons. First of all, we did not include the LR=0.0 for the choice of the hyperparameter. The hyperparameter range is exactly the same as that of TENT, reported in the original manuscript (P.7). Note that we separate the data for evaluation and validation even for the Test-domain validation, as the original paper suggests. Therefore, there is no guarantee that the LR=0.0 will be selected even if it is the best for the evaluation dataset.
> > >
> > > Secondly, we observed that the performance of SHOT is very unstable in several datasets (VLCS and TerraIncognita). In contrast, we can observe slight performance gain on the other two datasets (PACS and OfficeHome). These observations are common among the most optimization-based methods. The logic behind the observation needs more investigation, yet we suspect the label distribution difference (p(y|d)) causes the issue for the optimization-based approach. More specifically, we noticed that the label distribution is severely different among domains in VLCS and TerraIncognita compared to PACS and OfficeHome. Since the label difference is not considered in the loss function of existing methods, the optimization may cause failure in the target domain.
> > >
> > >
> > > > "The common practice in SFDA (offline setup) does not work well in an online setup" —what is the common practice in SFDA (updating most parameters?)? And offline vs. online here refers to multiple vs. single pass through the test data, correct?
> > >
> > > We are sorry for confusing you. Regarding the first question, we use the term to represents that the SFDA method updates most parameters, especially feature extractor. Regarding the online/offline thing, the distinction is correct. More specifically, the offline setting clearly separates adaptation and testing, yet these two phases are not distinguished in the online setting.
> > >
> > > ---
> > > Again, we thank the reviewer for reevaluating our works. We will add the results and discussion in the final manuscript and are happy to hear from you if you have further concerns.

---

> > > > ### Comment · Reviewer_mpea · 2021-08-29
> > > > **Response to authors**
> > > >
> > > > Dear authors,
> > > >
> > > > Thank you again for your detailed response – it addressed my concerns. I think you need to clearly mention the above reasons for the degraded SHOT performance in the offline test-domain-validation case – no label smoothing (key to the method) and a hyperparameter range that was designed for TENT, not SHOT. In the other cases, I think the benefits of your optimization-free method are clear.

---

### Official Review · Reviewer_3ToC · 2021-07-17

**Rating:** 6
**Confidence:** 4

**Summary:**

This paper introduces one algorithm for domain generalization and test-time template adjuster (T3A) for developing a model that performs well under conditions that are different from training conditions. The designed T3A is composed with two procedures: (1) compute a pseudo prototype representation for each class based on unlabeled online dataset, then (2) classify the data based on its distance to the pseudo prototypes. The proposed T3A can improve the performance on the unseen domains, and also outperform the state-of-the-arts.

**Limitations And Societal Impact:**

- The overall algorithm utilizes meta learning for boosting the transferability between the source and target domain. But based on the current experiment, the proposed method can perform well on the close-set dataset. I’m wondering if this method can perform well on the partial domain adaptation.
- The paper also mentions that the template trained by source data can’t guarantee that it is a good template for the target dataset. I’m also wondering to what extent that the template on the source data can reflect the feature of the transferred information?
- The format of section 5 is wired. If there is only one subsection, it doesn’t have to be one subsection.


**Main Review:**

Pros:
- Currently, domain generalization is a hot and important topic. Many works only focus on the training phase. However, T3A focuses on altering the testing phase based on the existing model, which is practical in the realistic world.
- The proposed method in algorithm 1 is simple and easy to implement.
- There is no training on feature extractor parameters, so the training is so fast.

Cons:
- In the proposed method, the feature extractors trained in each domain are fixed, and the goal is to solve a semi-supervised problem on a sample/prototype basis. Although not in the context of domain generalization, the reviewer thinks that this kind of problem setting was discussed before deep learning became popular. See, for example, reference [a].
-- [a] Driessens K., Reutemann P., Pfahringer B., Leschi C. Using Weighted Nearest Neighbor to Benefit from Unlabeled Data. PAKDD, 2006.
- A survey of sample/prototype-based semi-supervised learning methods prior to deep learning and a discussion of the superiority over these methods is needed.
- Except for the context of domain generalization, the novelty of the proposed method seems to be unclear.


**Time Spent Reviewing:**

3

---

> ### Author Response · Authors · 2021-08-10
> **Reply to Reviewer 3ToC**
>
> We thank the reviewer for the valuable feedback. Please find our responses below.
>
> ---
>
> **Difference with sample/prototype-based semi-supervised learning (SSL) and novelty of our paper.**
> Thanks for providing the missing literature. We agree that there is literature on sample/prototype-based methods, including literature on SSL (raised by reviewer 3ToC) and incremental learning (raised by reviewer rrrY). We will add the discussion on this point in the final manuscripts.
> Speaking of SSL, a primal difference between our setup and SSL is that the labeled data is truly reliable or not due to the existence of the distributional shift between labeled data and unlabeled data. This is why there is a massive amount of literature on unsupervised domain adaptation (UDA) setup, although the UDA is essentially the same with SSL except for the distributional differences. From a methodological perspective, our method does not use labeled source data to create templates, while the labeled data is usually used in the SSL setup.
>
> Second, and equally importantly, usually SSL is an offline setting where unlabeled data is available simultaneously as labeled data, but our setting is an online setting where unsupervised data is only available online. The distinction is essential since the deployed model usually needs to make correct predictions at that moment; there is no point in going back in time and correcting the predictions. In addition, our additional experiment (suggested by other two reviewers) clarifies the difference between online adaptation and offline adaptation. Specifically, just because a method works well in an offline setting does not mean that it works well in a more difficult online setting (see reply for reviewer mpea for more detail). The adaptation speed is also important in an online setup, as discussed in the current manuscript.
> In summary, our novelty is mixing the prototype techniques and pseudo label techniques and showing that it effectively handles the distributional differences even in the online adaptation setup. We hope this reply clarifies the novelty of this work.
>
> **Open-set experiments.**
> Thank you for providing an interesting setup. We agree that our method can be applied to various settings, and this is one of the potential merits of the proposed method. However, this paper primarily focuses on the standard domain generalization setup, and application to other setups is out of this paper's scope. We focus on the DG setup in this paper because the paper is primarily motivated by the recent work [16] that shows various DG algorithms fail to give significant performance gain compared to the simple ERM. Therefore, we try to provide new insight to this community by testing an unexplored approach in this field (focus on the test phase rather than the training phase). As all reviewers agree, empirical validations on multiple standard DG datasets clearly show that our approach is promising to increase performance stably.
>
> Speaking about partial domain adaptation setup, to the best of our knowledge, the main technical challenge tackled in the partial domain adaptation setup is how to avoid the miss-alignment of feature distributions between source and target due to the shared label space assumption [a, b]. We think this is not the case for our method since our method does not align feature distributions. We, therefore, think our method can be applied to partial domain transfer setup in theory, and there is no reason why the proposed method does not work well, while we need additional experiments to say for sure, which is outside the scope of this paper.
> Anyway, we thank the reviewer for giving an interesting setup. We agree that it is good to discuss the applicability of our method on a more realistic setup such as the open-set setup. We will add the above discussion to the final manuscript and clarify the primal scope of the paper more (in section 5).
>
> **The performance with templates on the source domains.**
> We are afraid we are not sure of the meaning of this comment. Assume N is the dimension of feature and K is the number of classes. Since the N by K weight matrices of the last linear layer can be naturally considered as the templates of each class on the source domains, the gap between w/o T3A and w/ T3A represents the merits of using templates on a target domain rather than directly using the templates on source domains. If you have further questions or concerns on this point, we are very welcome to hear that.
>
> **Format of section 5.**
> Thank you for giving a detailed comment. We will change the format in the final version.
>
> ---
> Again, thank you for giving us the opportunity to strengthen our manuscript with your valuable comments and queries. We will modify the manuscript following the above discussion.

---

### Author Response · Authors · 2021-08-10
**Results of Additional Experiments**

**Additional baselines**

Following reviewers' comments, we have conducted additional experiments to clarify the merits of the proposed method. Specifically, in addition to T3A, Tent-BN, and Tent-C, we tested the following six baselines.

- Tent-Full updates BN statistics and transformations.
- Norm [a] updates BN statistics but fixes transformations parameters.
- Pseudo label (PL) [b] updates entire networks by minimizing the cross-entropy between prediction and the pseudo label. Following [47], we assign the pseudo label if the predictions are over a threshold (0.9 in our experiment).
- PL-C updates the **linear classifier** by minimizing the above-mentioned pseudo-label loss.
- SHOT [32] updates feature extractor to minimize entropy, diversity regularizer, and pseudo-label loss.
- SHOT-IM [32] updates the feature extractor to minimize entropy and the diversity regularizer.



The table below summarizes methods and their characteristics.


|Method   |Optimization-free|Model-Agnostic|Updates                    |Hyperparameter                                     |Loss                                                         |
|---------|-----------------|--------------|---------------------------|---------------------------------------------------|-------------------------------------------------------------|
|SHOT-IM  |                 |✓             |feature extractor          |optimizer                                          |entropy, and diversity regularization                   |
|SHOT     |                 |✓             |feature extractor          |optimizer, threshold for PL, and balancing hyper parameter|entropy, diversity reguralizer, and cross entropy with pseudo label|
|PL       |                 |✓             |feature extractor, and classifier|optimizer, and threshold for PL               |cross entropy with pseudo label                              |
|PL-C     |                 |✓             |classifier                 |optimizer, and threshold for PL               |cross entropy with pseudo label                              |
|Tent-BN  |                 |✓             |added BN                   |optimizer                                          |entropy                                                      |
|Tent-C   |                 |✓             |classifier                 |optimizer                                          |entropy                                                      |
|Tent-Full|                 |              |BN transforms, and BN statics|optimizer                                          |entropy                                                      |
|Norm     |✓                |              |BN statics                 |/                                                  |/                                                            |
|T3A      |✓                |✓             |classifier                 |#supports                                        |/                                                            |


Here, we used resnet18 w/BN and resnet50 w/BN as backbone networks. Note that resnet18 and resnet50 used in the current manuscripts do not use the BN layer since it is the default option in the prior DG setup [16] (see appendix D of [16]). Therefore, please note that the results below are not directly comparable to the DG method shown in Table 1 of the current manuscript.

The following table summarizes the results when selecting the hyper-parameter in the training-domain validation set as with the current manuscript.

**Resnet18 w/ BN (training-domain validation)**

| Adaptation   | VLCS         | PACS         | OfficeHome   | TerraIncognita   |   Avg |
|:-------------|:-------------|:-------------|:-------------|:-----------------|------:|
| None         | 73.0 +/- 0.6 | 79.5 +/- 0.4 | 61.8 +/- 0.3 | 41.7 +/- 0.9     |  64   |
| SHOT-IM       | 61.6 +/- 0.3 | 82.1 +/- 0.3 | 62.8 +/- 0.2 | 32.8 +/- 0.4     |  59.8 |
| SHOT        | 61.8 +/- 0.3 | 82.3 +/- 0.2 | 62.8 +/- 0.2 | 32.7 +/- 0.4     |  59.9 |
| PL  | 67.0 +/- 0.6 | 72.9 +/- 1.0 | 56.3 +/- 2.5 | 35.4 +/- 1.7     |  57.9 |
| PL-C        | 71.8 +/- 1.3 | 78.9 +/- 0.4 | 61.7 +/- 0.3 | 43.1 +/- 0.9     |  63.9 |
| Tent-Full     | 72.3 +/- 0.3 | 83.9 +/- 0.3 | 62.7 +/- 0.2 | 36.9 +/- 0.3     |  64   |
| Norm     | 70.4 +/- 1.0 | 82.7 +/- 0.1 | 62.0 +/- 0.1 | 36.4 +/- 0.2     |  62.9 |
| Tent-C      | 71.3 +/- 1.5 | 74.6 +/- 1.9 | 60.5 +/- 0.4 | 40.9 +/- 0.5     |  61.8 |
| Tent-BN    | 64.7 +/- 0.7 | 81.1 +/- 0.2 | 62.5 +/- 0.3 | 36.4 +/- 0.9     |  61.2 |
| T3A          | 74.5 +/- 0.9 | 81.4 +/- 0.2 | 63.2 +/- 0.4 | 39.5 +/- 0.3     |  64.6 |

**Resnet50 w/ BN (training-domain validation)**


| Adaptation   | VLCS         | PACS         | OfficeHome   | TerraIncognita   |   Avg |
|:-------------|:-------------|:-------------|:-------------|:-----------------|------:|
| None         | 74.3 +/- 0.5 | 84.1 +/- 0.1 | 66.9 +/- 0.2 | 45.8 +/- 1.8     |  67.8 |
| SHOT-IM       | 61.5 +/- 1.7 | 84.8 +/- 0.4 | 68.0 +/- 0.0 | 33.8 +/- 0.3     |  62   |
| SHOT         | 61.6 +/- 1.8 | 84.8 +/- 0.5 | 68.0 +/- 0.0 | 34.6 +/- 0.3     |  62.3 |
| PL  | 63.4 +/- 1.8 | 80.1 +/- 3.5 | 61.3 +/- 1.5 | 36.8 +/- 4.4     |  60.4 |
| PL-C        | 73.3 +/- 0.8 | 84.7 +/- 0.3 | 66.4 +/- 0.3 | 47.0 +/- 1.7     |  67.9 |
| Tent-Full     | 75.4 +/- 0.6 | 87.0 +/- 0.2 | 66.9 +/- 0.2 | 42.6 +/- 0.8     |  68   |
| Norm     | 71.3 +/- 0.4 | 85.8 +/- 0.1 | 66.4 +/- 0.1 | 42.3 +/- 0.4     |  66.5 |
| Tent-C      | 72.4 +/- 1.5 | 84.4 +/- 0.1 | 66.2 +/- 0.2 | 42.4 +/- 3.1     |  66.4 |
| Tent-BN    | 65.6 +/- 1.4 | 84.9 +/- 0.0 | 67.7 +/- 0.2 | 42.7 +/- 0.5     |  65.2 |
| T3A          | 76.0 +/- 0.3 | 85.1 +/- 0.2 | 68.2 +/- 0.1 | 44.6 +/- 0.9     |  68.5 |


In addition, the following table summarizes the results when selecting the hyper-parameter in the test-domain validation set (Oracle setup) as suggested by the reviewer2. Specifically, we chose hyper-parameters using the validation set on the target domain (20% of all data as described in the manuscript). Note that this is a bit unrealistic setup since it is usually better to use the labeled data to construct the model if it is available. For example, our method can naturally use the labeled data to create more reliable templates.

**Resnet18 w/ BN (test-domain validation)**

| Adaptation   | VLCS         | PACS         | OfficeHome   | TerraIncognita   |   Avg |
|:-------------|:-------------|:-------------|:-------------|:-----------------|------:|
| None         | 73.0 +/- 0.6 | 79.5 +/- 0.4 | 61.8 +/- 0.3 | 41.7 +/- 0.9     |  64   |
| SHOT-IM       | 60.9 +/- 0.3 | 81.9 +/- 0.3 | 62.4 +/- 0.3 | 30.5 +/- 0.4     |  58.9 |
| SHOT         | 61.2 +/- 0.4 | 81.4 +/- 0.2 | 62.4 +/- 0.3 | 32.3 +/- 0.6     |  59.3 |
| PL  | 65.4 +/- 0.2 | 73.1 +/- 2.5 | 54.7 +/- 4.6 | 37.7 +/- 2.5     |  57.7 |
| PL-C        | 72.5 +/- 1.3 | 78.4 +/- 0.8 | 61.8 +/- 0.3 | 43.9 +/- 1.3     |  64.2 |
| Tent-Full     | 73.8 +/- 0.8 | 84.7 +/- 0.2 | 62.7 +/- 0.1 | 37.4 +/- 0.8     |  64.7 |
| Norm     | 70.3 +/- 0.9 | 82.7 +/- 0.1 | 62.1 +/- 0.1 | 36.6 +/- 0.2     |  62.9 |
| Tent-C      | 72.3 +/- 1.1 | 77.3 +/- 1.8 | 61.3 +/- 0.3 | 37.8 +/- 2.7     |  62.2 |
| Tent-BN    | 64.7 +/- 0.7 | 81.2 +/- 0.2 | 62.6 +/- 0.3 | 36.4 +/- 1.0     |  61.2 |
| T3A          | 73.8 +/- 0.8 | 81.3 +/- 0.0 | 62.8 +/- 0.3 | 40.5 +/- 0.3     |  64.6 |

**Resnet50 w/ BN (test-domain validation)**

| Adaptation   | VLCS         | PACS         | OfficeHome   | TerraIncognita   |   Avg |
|:-------------|:-------------|:-------------|:-------------|:-----------------|------:|
| None         | 74.3 +/- 0.5 | 84.1 +/- 0.1 | 66.9 +/- 0.2 | 45.8 +/- 1.8     |  67.8 |
| SHOT-IM       | 59.6 +/- 2.7 | 84.7 +/- 0.4 | 67.5 +/- 0.3 | 27.0 +/- 1.4     |  59.7 |
| SHOT         | 61.4 +/- 1.5 | 85.1 +/- 0.4 | 67.5 +/- 0.3 | 30.9 +/- 2.3     |  61.2 |
| PL  | 64.2 +/- 2.3 | 70.4 +/- 1.0 | 55.0 +/- 1.8 | 33.0 +/- 7.4     |  55.6 |
| PL-C        | 73.5 +/- 0.9 | 84.6 +/- 0.4 | 66.7 +/- 0.2 | 46.7 +/- 1.6     |  67.9 |
| Tent-Full     | 74.8 +/- 1.2 | 87.7 +/- 0.1 | 67.0 +/- 0.4 | 42.9 +/- 0.2     |  68.1 |
| Norm     | 71.4 +/- 0.5 | 85.7 +/- 0.1 | 66.6 +/- 0.0 | 42.4 +/- 0.4     |  66.5 |
| Tent-C      | 72.6 +/- 0.8 | 83.3 +/- 0.5 | 66.7 +/- 0.2 | 45.0 +/- 0.9     |  66.9 |
| Tent-BN    | 65.7 +/- 1.4 | 84.9 +/- 0.0 | 67.7 +/- 0.2 | 42.7 +/- 0.5     |  65.3 |
| T3A          | 76.1 +/- 0.2 | 85.2 +/- 0.2 | 67.8 +/- 0.2 | 46.0 +/- 1.5     |  68.8 |



We can make the following observations.
1. When we select the hyperparameters with the training-domain validation set, the proposed method still outperformed all baselines in both backbone networks. Among baselines, only Tent-Full and PL-C perform better than None (w/o adaptation) on average.
2. When we select the hyperparameters with a test-domain validation set, Tent-Full gives comparable performances with the proposed method. In addition, compared to T3A and PL-C, the T3A performs better under both model selection strategies. These results clarify the difficulty of model selection in optimization-based methods and the merit of the proposed optimization-free approach in this setup.
3. Updating feature extractor (or the large portion of the parameters) does not work well in general, while it is common in SFDA (offline) setup. The results suggest that we need different treatments on online and offline setup.

Again, we thank the reviewers for providing detailed comments. We will add the above results to the final manuscript.


References
- [a] Schneider, Steffen et al. “Improving robustness against common corruptions by covariate shift adaptation.” NeurIPS (2020)
- [b] Lee, Dong-Hyun. “Pseudo-Label : The Simple and Efficient Semi-Supervised Learning Method for Deep Neural Networks.” ICML Workshop (2013).

---

### Decision · Program_Chairs · 2021-09-27

**Decision:**

Accept (Spotlight)

**Comment:**

The paper is proposing a test-time adaptation method for domain generalization. The proposed idea is quite simple and effective. It uses the predictions as pseudo labels and only updates the final classification layer without any gradient-based optimization. Hence, it is not only simple and effective but also lightweight. All reviewers agreed that the proposed method has merits; however, there were some concerns with the empirical study. The authors provided significant additional empirical support during rebuttal and reviewers appreciated it. I believe it is a solid paper that will likely have a significant impact on the community.